# A novel causal structure-based framework for comparing basin-wide water-energy-food-ecology nexuses applied to the data-limited Amu Darya and Syr Darya river basins

Haiyang Shi[1,2,4,5], Geping Luo[1,2,3,5], Hongwei Zheng[1], Chunbo Chen[1], Olaf Hellwich[6], Jie Bai[1], Tie Liu[1,5], Shuang Liu[1], Jie Xue[1], Peng Cai[1,4], Huili He[1,4], Friday Uchenna Ochege[1,2], Tim van de Voorde[4,5], and Phillipe de Maeyer[1,2,4,5]

[1] State Key Laboratory of Desert and Oasis Ecology, Xinjiang Institute of Ecology and Geography, Chinese Academy of Sciences, Urumqi, Xinjiang, 830011, China.
[2] University of Chinese Academy of Sciences, 19 (A) Yuquan Road, Beijing, 100049, China.
[3] Research Centre for Ecology and Environment of Central Asia, Chinese Academy of Sciences, Urumqi, China.
[4] Department of Geography, Ghent University, Ghent 9000, Belgium.
[5] Sino-Belgian Joint Laboratory of Geo-Information, Ghent and Urumqi.
[6] Department of Computer Vision & Remote Sensing, Technische Universität Berlin, 10587 Berlin, Germany

*Correspondence to*: Geping Luo (luogp@ms.xjb.ac.cn)

Submitted to *Hydrology and Earth System Sciences*

Special issue: Socio-hydrology and Transboundary Rivers

**Abstract.** The previous comparative studies on watersheds were mostly based on the comparison of dispersive characteristics, which lacked systemicity and causality. We proposed a causal structure-based framework for basin comparison based on the Bayesian network (BN), and focus on the basin-scale water-energy-food-ecology (WEFE) nexuses. We applied it to the Syr Darya river basin (SDB) and the Amu Darya river basin (ADB) of which the poor water management caused the Aral Sea disaster. The causality of the nexuses was effectively compared and universality of this framework was discussed. In terms of changes of the nexuses, the sensitive factor for the water supplied to the Aral Sea changed from the agricultural development during the Soviet Union period to the disputes in the WEFE nexuses after the disintegration. The water-energy contradiction of SDB is more severe than that of ADB partly due to the higher upstream reservoir interception capacity. It further made management of the winter surplus water downstream of SDB more controversial. Due to this, the water-food-ecology conflict between downstream countries may escalate and turn into a long-term chronic problem. Reducing water inflow to depressions and improving the planting structure prove beneficial to the Aral Sea ecology and this effect of SDB is more significant. The construction of reservoirs on the Panj river of the upstream ADB should be cautious to avoid an intense water-energy conflict as SDB. It is also necessary to promote the water-saving drip irrigation and to strengthen the cooperation.

## 1 Introduction

The Aral Sea disaster has warned us for the terrible impact of unsustainable water use on the ecosystem. Recently, with the growing focus on the water-energy-food (WEF) nexus (Biggs et al., 2015; Cai et al., 2018; Conway et al., 2015; Espinosa-Tasón et al., 2020; Sadeghi et al., 2020; Yang and Wi, 2018) in the integrated water resources' management, we have come to realize that a harmonious and optimized water-energy-food-ecology (WEFE) nexus may be the key to an effective cross-border water management of the Aral Sea basin (Jalilov et al., 2016, 2018; Lee and Jung, 2018; Ma et al., 2020; Sun et al., 2019), with 'ecology' added to the WEF nexus because ecology is usually more concerned in the Aral Sea basin. The latter mainly includes the Syr Darya river basin (SDB) and the Amu Darya river basin (ADB). Due to the similarity in the natural geographical conditions and management approaches, these two basins are generally considered to be very similar. The rapid melting of glaciers, drought disasters, excessive irrigation water use, increasing food demand, contradictions on water for the energy production and irrigation between the upstream and downstream countries, soil salinization and poor water quality are the common problems the two basins are facing nowadays (Immerzeel et al., 2020; Micklin, 2010). However, there seems to be a lack of attention to the quantitative differences on the characteristics of the interactions of the WEFE nexus between the two river basins. We want to understand the differences and their levels, and think about what experience can be gained from it. The practice of an integrated watershed management often draws on the experience and lessons of other watersheds with similar natural conditions, such as management concepts, hydrological model applications and climate change risk assessments (Grafton et al., 2012; Immerzeel et al., 2020; Joetzjer et al., 2013; Ladson and Argent, 2002; Syed et al., 2005; Vetter et al., 2017; Wang et al., 2020; Zawahri, 2008). Most of these previous studies investigated the differences of dispersive or individual characteristics between the river basins but lacked attention to the systemicity and causality (Fig. 1) in the changing water systems at the basin scale which may be able to more directly provide new experience

and knowledge for practical watershed management. In SDB and ADB, this kind of comparison might be more practical and meaningful on the application level (based on a higher similarity in the natural conditions and management history). Learning from each other's successes and failures could reduce the trial-and-error costs in the water use management. For example, the seasonal runoff pattern and its impact on the water use of SDB nowadays with a low glacier cover might be considered as a reference for the water use management of ADB, if most glaciers would melt in a warmer future (Sorg et al., 2012). Analogously, such comparisons are focusing on the detailed differences under a general similarity and might also be helpful to understand the WEFE nexus and a better assignment of the detailed responsibilities of countries regarding a transboundary watershed cooperation and management.

When studying the water system and the WEFE nexus in the Aral Sea basin, we found that the first main source of uncertainty might include the fact that it is difficult for us to accurately predict the runoff amount from the mountainous areas. In the arid regions of Central Asia, most of the available water resources originate from the precipitation, melting snow and glaciers of the water towers in the alpines. But the observations of the water resources in the mountainous areas of this region have been greatly restricted (Chen et al., 2017), especially after the collapse of the Union of Soviet Socialist Republics (USSR) and some gauging stations were abandoned. It has restricted the implementation of the physics-based and statistical models for the runoff prediction, although remote sensing technology proved helpful in the estimation of the alpine precipitation and glacier melting (Guo et al., 2017; Pohl et al., 2017) as forcing data. In addition, the weak prediction capacity of incoming water might propagate the uncertainty on the downstream water use, food production, energy production, ecology and their interactions in the WEFE nexus. Facing the uncertainty of the amount of incoming water and some other exogenous sources such as climate change and population growth, some models concerning the WEF nexus that are commonly used now, may not work well. Previous studies focused more on the WEF nexus in the integrated water resources' management (IWRM) (Cai et al., 2018) and many current WEF nexus studies applied the system analysis or integrated process-based model methods (Daher and Mohtar, 2015; Jalilov et al., 2018; Kaddoura and El Khatib, 2017; Lee et al., 2019, 2020; Payet-Burin et al., 2019; Zhang and Vesselinov, 2017). However, in order to parameterize these models, we found that many empirical parameters or factors need to be set (Feng et al., 2016; Ravar et al., 2020), which could mask the shortcomings of an insufficient understanding of uncertain and complex processes. For example, empirical coefficients were used to determine the conversion coefficient of electricity demand for pumping water from different depths and energy demand coefficients of various water sectors (Ravar et al., 2020), including the driving functions of water supply, power generation and hydro-ecology (Feng et al., 2016). The effectiveness depends on our judgements of the values of each parameter under various conditions, but we might ignore the dynamic influence of the probability distribution of some remotely related causal variables. In order to improve this, we considered a longer causal chain matching of the uncertainty propagation process and to obtain details on the possibility distributions of the parameters' values under various combinations of multiple conditions. Therefore, we realized that the Bayesian network might prove to be an effective tool for these two problems.

The Bayesian network (BN) is based on the Bayesian theory and the graph theory (Friedman et al., 1997; Pearl, 1985). It can simulate complex causal relationships and integrate expert knowledge from multiple fields and has shown its advantages in water resources research and management (Chan et al., 2010; Fienen et al., 2013; Giordano et al., 2013;

Hines and Landis, 2014; Hunter et al., 2011; Nash and Hannah, 2011; Pagano et al., 2014; Quinn et al., 2013; Taner et
al., 2019; Xue et al., 2017). In our previous study, the WEFE nexus in the single SDB was simulated based on a BN
(Shi et al., 2020) which also demonstrated its advantages in terms of uncertainty quantification. Based on this, we try
to explore the framework significance and portability of this method when applied to other watersheds for comparing
watershed systemic behaviours focusing more on the global causality, which aimed at obtaining the universal evolution
law and discovering the specific differences of the basin-wide WEFE nexus.
The research goals of this paper mainly include: (1) to propose a causal structure-based framework to compare basin-
wide WEFE nexuses and apply it to SDB and ADB with the BN method, (2) to compare the differences in historical
and current causality of the WEFE nexus and water use between SDB and ADB within the new framework and (3) to
propose a comprehensive optimization proposal of the WEFE nexus management.
**2 A generalized causal structure-based framework for comparing basin-wide water-energy-food-ecology**
**nexuses**
We propose a new framework (Fig. 2) for comparing the basin-wide WEFE nexuses and watershed management
representing the causal structure based on combining the similar causal structure and data differences. Under different
levels of similarity, similar causal structures generated by expert knowledge are combined with the observation and
statistical datasets of different river basins. The elements of the WEFE nexus can be adjusted to water-energy, water-
food-ecological nexus (Fig. 2), etc. according to the dynamic research aims and similarity levels among the specifically
investigated river basins.
The steps of the workflow of the framework are as follows:
(1) We conduct a preliminary screening of the basin. Such screening can be based on similar geographic region,
landform, climate type, etc. which reflect the basic natural conditions. Based on other factors such as whether the river
is transboundary, whether the country that manages the basin is economically developed, etc., we further filter the
selected basins.
(2) We construct a same WEFE nexus causality structure for the river basins selected in the previous step, which can
be represented by a directed graph model such as the Bayesian network. In this step, we need to balance the degree of
refinement of the causal relationship structure and its universality in the selected river basins. The conceptual structure
constructed should be reviewed by a panel of experts and revised if necessary. This feedback can help to identify key
variables or processes that have been overlooked so as to correct errors in the conceptual structure. In some cases, it
may be appropriate to build a conceptual structure with stakeholder groups, especially if the model will be used as a
management tool and the results will affect stakeholders (Chan et al., 2010; Chen and Pollino, 2012). At the same time,
the availability of actual expert knowledge and data should also be considered to avoid constructing a causal structure
that is too detailed so that the available expert knowledge and data are not enough to fill it, or too rough that the causal
relationship is underfitted so as to avoid underutilization of knowledge and data (Chen and Pollino, 2012; Marcot et
al., 2006). Including insignificant variables will increase the complexity of the network and reduce the sensitivity of
the model output to important variables, unnecessarily spending extra time and effort, and will not add value to the
entire model (Chen and Pollino, 2012).
(3) In this step, we combine the causal structure representing expert knowledge from multiple fields with actual
statistics and observation data to update the initial understanding of causality (Cain, 2001; Chan et al., 2010; Chen and
Pollino, 2012; Marcot et al., 2006). Expert judgment based on past observations, knowledge and experience can be
used to provide an initial estimate of the probability, which can then be updated with the available observation data
(Chen and Pollino, 2012). The ability to use expert opinions to parameterize the BN model is an advantage, especially
for environmental systems that have little quantitative data required for statistical modeling methods (Smith et al.,
2007). In this way, the conditional probability table of the original causal structure is updated with actual data, and the
originally scattered actual data is closely connected by the causal structure.
(4) Based on the quantified new causal structure in the previous step, we can explore its value in practical applications
within the new framework including: discovering the common evolutionary law of the nexuses, discovering the
differences in the responses of various nodes to the same management scenario by synchronizing the operations of
BNs of different river basins, analyzing differences of the causality of the historical nexuses changes, incorporating
previous unsystematic and local studies on water resources, agriculture, ecology, etc. into the new causal framework
such as incorporating the upstream multi-source causal factors into the downstream soil salinization studies, sharing
experience and reflecting on the failure cases of the historical management, optimizing the current nexuses,
incorporating causality and uncertainty into the decision making and the future risk assessment (Chan et al., 2010).
**3 Application of the Framework in the Syr Darya river basin (SDB) and the Amu Darya river basin (ADB)**
**3.1 Location of the selected SDB and ADB**
The Aral Sea Basin is located in Central Asia (Fig. 3) with a total area of 1,549 million km$^2$ and is one of the largest
endorheic river basins in the arid regions worldwide. The two major rivers, the Syr Darya and the Amu Darya, originate
from the West Tien-Shan and Pamir Plateau as a part of the Central Asian water tower. They flow through five
countries in Central Asia, which were once part of the USSR. The surface water resources of the basin mainly stem
from the precipitation, snow melting and ice in the mountainous area. The lower part of the basin is very dry and most
areas are deserts. The large-scale agricultural production here is highly dependent on the irrigation and large amounts
of water are consumed by a high evapotranspiration and leakage during the water diversion.
**3.2 The priori and general mode of the water-energy-food-ecology (WEFE) nexus of SDB and ADB**
Since the 1960s, the WEFE nexus in the Aral Sea Basin has been suffering from an increasing pressure (Fig. 4). In
addition to the population growth, climate change, ecological degradation and other problems, the issue of the
transboundary water and energy disputes in this region has intensified with the collapse of the USSR. Therefore, this
basin-wide transboundary WEFE nexus has unique characteristics on spatial and chronological scales. In this study,
according to the spatial characteristics of the transboundary management, the watershed is divided into an upstream
and downstream area. In response to the impact of the collapse of the USSR, the water resources' management period
was divided into four periods: namely 1970-1980, 1980-1991, 1991-2005 and 2005-2015. This is mainly based on the
WEFE nexus change between the upstream and downstream areas in different periods, which are applicable to both
SDB and ADB as a priori and general mode:
(1) The agricultural development stage (1970-1980): During this period, a large-scale land development was carried
out, mainly planting cotton with high water consumption and by means of flood irrigation. During this period, large-
scale reservoirs, irrigation and drainage canals and other hydraulic irrigation projects were built. With serious leakage
and a low efficiency, a large amount of water resources was being consumed before going to the farmlands and the
water amount entering the Aral Sea has already begun to decrease (Micklin, 1988).
(2) Cultivated land development reaches the highest level and agricultural production continued to be high-load (1980-
1991): During this period, because the Aral Sea basin was regarded as the main agricultural production area of the
USSR, the agricultural demand was extremely large. When the agricultural products were ready, they were handed
over to Moscow, where they were uniformly distributed to other regions of the USSR. The scale of the agricultural
development has reached its peak and was relatively stable. The water amount entering the lake from the Aral Sea has
been reduced further (Micklin, 2007, 2010). In some years, even river depletion occurred. The agricultural water in the
downstream area was given priority and the gap in the upstream power generation needs was compensated for by free
fossil energy from the downstream area. The operation mode of the reservoir in the upstream mountain area was close
to the natural mode. When the summer streamflow was large, the reservoir outflow was also high in order to ensure
the agricultural water use in the lower part.
(3) The stage of economic stagnation after the collapse of the USSR (1991-2005): The politic in the newly born Central
Asian countries remained unstable during this period and there was a social and economic stagnation. The cotton
production scale of the previous USSR period was far greater than the actual demand of the five new countries. The
area of agricultural land has decreased. But due to population growth and the new countries' own food security needs,
the proportion of food crops grown has increased. The downstream area no longer supplied energy to the upstream
area for free. The upstream region had an energy crisis and the demand for electricity was not met, especially in the
cold winter during the peak in electricity consumption. In order to ensure the electricity supply in winter, the upstream
countries increased the interception water with reservoirs in the high mountains during summer and released more
water in winter so as to generate electricity. This resulted in a downstream agricultural water shortage in summer and
flood risk during winter (Micklin, 2007, 2010). The long-term flood irrigation has caused serious salinization and
decreased the fertility of the farmland soil downstream. Pesticides and salt in the return flow of irrigation entered the
river, causing the downstream water quality to decline. The exposed Aral Sea lake bed increased the frequency of the
sand and salt dust storms, threatening the health of the residents and the Aral Sea crisis developed further as a result.
(4) The stage of socioeconomic recovery (2005-2015): Kazakhstan and Turkmenistan were rich in fossil energy and
have a certain foundation for industrial development, have experienced a rapid economic development. Relatively
wealthy, Kazakhstan built large reservoirs so as to prevent floods and to regulate the irrigation, alleviating its own
disadvantages in the water resources' competition. Turkmenistan withdraws more water, along with the economic
development and population growth. The energy disputes between the upstream and downstream areas have become
increasingly fierce. For example, the amount of natural gas exported from Uzbekistan to the upstream region, was
greatly reduced. The power satisfaction and living standards of the upstream countries have only improved little. The
Aral Sea continued to shrink and by 2010, only 10 % of the area was left compared to the 1960s (Micklin, 2010).

**3.3 A general Bayesian network (BN) structure with macro spatial information within the new framework applied to SDB and ADB**

We separated the upstream area, downstream area and the Aral Sea as geographically discrete regions and introduced
the elements in the WEFE nexus joint to these regions into the BN as different variables (Fig. 5). Each variable
represents a certain element in the WEFE nexus of a certain region. The BN could be divided into six modules,
including the natural water resources, upstream, downstream, Aral Sea and target variables and a causal structure has
been established based on the expert experiences (Fig. 6). We established this common framework as a prerequisite
for establishing a joint probability table and at the same time we tried to adapt SDB and ADB so as to keep each
variable universal, although the specific meaning of the variables should be different in the two river basins. The
responsibility for exploring the differences between the two river basins mainly relies on the input observation data.

**3.4 Compiling and Evaluation of the BN**

A BN describes the joint probability distribution of the set of nodes. For a BN in which nodes represent random
variables ($X_1,.,X_n$), its joint probability distribution P(X) is given as (Pearl, 1985):

$$P(X) = P(X_1, X_2, \ldots, X_n) = \prod_{i=1}^{n} P(X_i|pa(X_i)) \tag{1}$$

where $pa(X_i)$ are the values of the parents of $X_i$ and $X_1,.,X_n$ are variables in the WEFE nexus structure. Based on the
expert knowledge, we initially gave values to the corresponding conditional probability table for each node of the BN.
We discretized the value range of nodes to reduce computational requirements (Table 1). The discretized interval also
has a certain extension to ensure the robustness of the later prediction function and to prevent cases from easily
exceeding the boundary. According to the differences in the political and economic backgrounds at different stages,
we divided the development process during the past 50 years into four stages: 1970-1980, 1980-1991, 1991-2005 and
2005-2015, based on the assumption that the WEFE nexus shows a relative stability under similar political and
economic backgrounds. Next, in order to integrate actual observations and statistical data, the expectation–
maximization (EM) algorithm (Moon, 1996) function of Netica software is used to iteratively calculate the joint
probability distribution of BN. In the Netica software, the "experience" variable is used to indicate the reliability of the
prior knowledge, and the "degree" variable is used to indicate the training times of the observation data. By combining
these two variables, we can dynamically adjust and balance the weights of prior knowledge and the actual data in the
probability distribution updation. In this study, we initially set "experience" <0.3 "degree" to ensure that the weight of
the information represented by the actual data is sufficient.
To assess the degree of agreement between the parameterized of BN and the actual situation, we used the sensitivity
analysis of the BN (Castillo et al., 1997; Laskey, 1995; Marcot, 2012). The index variance of belief (VB) and the index
mutual information (MI) based on the change of information entropy (Barton et al., 2008; Marcot, 2012) are applied
to evaluate the change in strength and uncertainty of the causal relation between the nodes. They respectively represent
the reduction in variance and entropy of the probability distribution of child nodes caused by the determination of the
state of the parent nodes. As the value range of the parent node is reduced, the variance or entropy of its distribution is
usually reduced. The greater the variance or entropy of the distribution of child nodes that can be further caused by
this reduction, the more sensitive the child node is to the parent node which also reflects the stronger causality. These
two indicators are as follows:
$$MI = H(Q)\text{-}H(Q|F) = \sum_q \sum_f P(q, f) \log_2 \left( \frac{P(q,f)}{P(q)P(f)} \right) \tag{2}$$

$$VB = V(Q)\text{-}V(Q|F) = \sum_q P(q)\left[X_q - \sum_q P(q)X_q\right]^2 - \sum_q P(q|f)\left[X_q - \sum_q P(q|f)X_q\right]^2 \tag{3}$$

where H stands for the entropy, V stands for the variance, Q stands for the target node, F stands for other nodes and q
and f stand for the status of Q and F. Xq is the true value of the status q.

**3.5 A BN-based analysis of the historical factors on the water entering the Aral Sea, the post-test probability**
**prediction and multi criteria evaluation with the Markov chain-Monte Carlo sampling**

We used the index VB that is utilized in the sensitivity analysis to analyze the factors that affect the water entering the
Aral Sea in the four stages during the past 50 years. It is mainly significant to form a quantified understanding that was
originally only qualitative. Quantifying and updating the past knowledge can help us to better understand the impact
and differences of the water resources' development and the WEFE nexus change at different stages in SDB and ADB.
Because the difference in the current status of the two rivers may have been accumulated from the historical differences
in the water-land-energy development during the past 50 years.
We utilized the posterior probability prediction function of BN so as to support the decision optimization. Assuming
that the values of some variables have been determined, the posterior probability prediction of BN might be employed
to infer the possible effect on the variables we are concerned about. The prediction function is usually used to infer
and predict how one node (D) is likely to change with the distribution of its parent node (A) determined. All nodes that
have dependencies between A and D should be included in the calculation. For example, suppose we have the simple
Bayesian network for discrete variables with the structure A and D are connected through a dependency of D on C, C
on B and B on A, and we can use the following formula (Heckerman and Breese, 1996) to calculate the probability of
D when the state of A is given.
$$P(D|A) = \frac{P(A, D)}{P(D)} = \frac{\sum_{B,C} P(A, B, C, D)}{\sum_{A,B,C} P(A, B, C, D)} = \frac{P(A) \sum_B P(B|A) \sum_C P(C|B)P(D|C)}{\sum_A P(A) \sum_B P(A)P(B|A) \sum_C P(C|B)P(D|C)} \tag{4}$$

Parent nodes are regarded as the independent variables, child nodes are regarded as the objectives. When the state of
parent node is given, the beneficial probability distribution change of the child node can be regarded as our optimization
goal. We formulated a change measure (ΔP) (Robertson et al., 2009; Xue et al., 2017) to assess the impact of a
management scenario compared to a base case:
$$\Delta P_{low} = P(X_i|e)_{low} - P(X_i)_{low} \tag{5}$$

$$\Delta P_{high} = P(X_i|e)_{high} - P(X_i)_{high} \tag{6}$$

where e represents the determination of the state of the parent node (management scenario) in the form of hard evidence
specifying a definite finding, $P(X_i|e)_{low}$ is the probability of the lowest state for the management scenario, $P(X_i)_{low}$ is
the probability of the lowest state for the base case and $\Delta P_{low}$ is calculated as the change. The meanings of these
variables are the same for the subscripts 'high'.

The goal of the above optimization only contains a single variable, to test whether they seemed beneficial under
multiple comprehensive criteria, we selected the scenarios with a good effect ('reducing the water inflow to the
depression' and 'improving the planting structure') for the multi-criteria (combination of the above single target
variables) assessment. Based on the Markov chain-Monte Carlo (MCMC) (Neal, 1993) sampling of the BN, we explore
its role in multi-criteria assessment and optimization based on previous studies (Farmani et al., 2009; Molina et al.,
2011; Shi et al., 2020; Watthayu and Peng, 2004). The point or solution set obtained from MCMC sampling matches
the high-dimensional joint probability distribution of BN nodes, which encompasses the causality of the system (Neal,
1993). This will be applied so as to determine the size of the uncertainty behind the optimization effect of the scenario
and to verify the ability of the BN to manipulate the multi-dimensional uncertainty in the decision-making. When the
states of some nodes in the BN are determined, the joint probability distribution of the posterior changes, and the
distribution of the point set in the multi-criteria space also changes accordingly. The distribution of this point set is
constrained by the causality constructed by BN. If the pareto solutions obtained by conventional system optimization
analysis are far outside the distribution range of this point set, then these optimization solutions may actually not meet
the true causality constraints as an overestimated optimized solution that does not conform to the reality. In addition,
this process could be seen as a test of the robustness of the optimization solutions. The degree in dispersion of the
optimization cases in the three-dimensional criterion space could visually illustrate the size of its uncertainty, which is
helpful for the decision- making with intuitively displaying a high-dimensional joint probability. The three indicators
the reliability (REL) (Cai et al., 2002), total benefit (TB) and degree of cooperation (DC) (Shi et al., 2020) used for
multi-criteria evaluation are as follows:
$$REL = \beta \frac{HA}{A} + (1 - \beta) \frac{WECO}{TWECO} \qquad (7)$$
where HA is the planted area, A represents the area suitable for planting, WECO determines the ecological flow
calculated as the water entering the Aral Sea, TWECO is the target flow and $0 \leq \beta \leq 1$ is an adjustable weight.
$$TB = P_a \times AP + P_e \times EB + P_h \times HP \qquad (8)$$
$$DC = HP/AP \qquad (9)$$
where HP indicates the benefits of hydroelectric power generation from upstream dams. EB is the benefit of
downstream ecological flow entering the Aral Sea which is calculated as a linear function of WECO in this paper. AP
indicates the agricultural production in downstream countries. $P_a$, $P_h$ and $P_e$ are the prices or weights which can be
adjusted according to the actual market price in the international trade when it comes to cross-border cooperative
management in which different types of benefits (such as upstream hydropower and downstream agricultural products)
may need to be weighted and summed. It may be more reasonable to use the universal price of various benefits in the
international market to determine the weight. The value of ecological flow can be calculated as the value of the
ecosystem services it provides. As a simplified calculation, we normalized the three indicators to 0-1 and sum them
with equal weights.

**3.6 Data**

We collected data on the WEFE nexus from 1970 to 2015 in the Aral Sea basin (Table 2). They will be entered into
the BN along with expert knowledge. For SDB, the upstream area includes Kyrgyzstan and the downstream area covers
Kyzylorda, Shymkent in Kazakhstan and Namangan, Andijan, Fergana, Jizzakh, Syrdarya and Tashkent in Uzbekistan.
Regarding ADB, the upstream region includes Tajikistan and the downstream region comprises Surxondaryo,
Qashqadaryo, Samarqand, Bukhara, Navoiy, Khorezm, Karakalpakstan in Uzbekistan and the entire Turkmenistan.

**4 Results**

**4.1. Model evaluation**

We input the collected data and expert knowledge into the BN and compiled it with the EM algorithm in the Netica.
In this study, we selected four nodes as target variables for a sensitivity analysis (Fig. 7). We found that VB and MI
have similar trends, and when VB is larger, MI is also larger. This indicates that the correlation and uncertainty between
nodes are synchronized in response to changes in the parent node. The upstream power generation of the two basins is
sensitive to the hydropower and imported energy. The downstream water use is more sensitive to agricultural water
and living water use. The downstream agricultural production is very sensitive to crop production, animal husbandry
production and soil salinization. The water inflow to the Aral Sea is sensitive to runoff, water use and reservoir
operation. The ranking of these sensitivity factors matches our knowledge and experience about the Aral Sea basin
well. Since the impact of the other variables in the BN gradually decreases as the number of intermediate variables
increases, these sensitivity results match well with expert and stakeholder perspectives. A strong pseudo-causality was
not found between two variables with no obvious prior causality. In general, the variables with a strong causality are
directly connected in the network. This indicates that the established priori causal structure has withstood the test of
the actual data.

**4.2 Comparing the WEFE nexus of SDB and ADB during the past 50 years**

We applied the sensitivity analysis to the node 'water inflow to the Aral Sea' of SDB and ADB at different historical
stages (Fig. 8). During the period 1970 - 1980, there was no significant difference between the influencing factors of
the two river basins and the related variables of the increased agricultural development contributed greatly. With the
completion of the upstream reservoirs, the rising reservoir storage also had a certain contribution in both river basins.
In this period, the variability of the natural runoff of the Syr Darya River was significantly larger than the Amu Darya
River's and the contribution of the natural runoff was higher. During the period 1980 - 1991, the contribution of most
variables has declined, which may be related to the normalization of the maximized agricultural production, leaving
only the natural runoff as the main variation contribution. During the period 1991 - 2005, for SDB, the contribution of
the water inflow into the depression has risen significantly. In both river basins, the reservoir storage and summer
release contribution also augmented largely, with SDB even higher, and the support of the upstream energy import
from the downstream area has also increased. During the period 2005 - 2015, for SDB, the contributions of the
agricultural water and downstream crop area has rosen significantly and the output of the water inflow to the depression
has been decreasing.

In general, before the collapse of the USSR, the difference was mainly sourced from the runoff variability and the
proportion of the upstream reservoir interception to the total natural runoff. The runoff proportion of the Naryn River
tributary (about 35% of the total runoff of the Syr Darya river) intercepted by the Toktogul hydropower station, was
higher than the one of the Vakhsh River tributary (about 25% of the total runoff of the Amu Darya river) intercepted
by the Nurek hydropower station. It also shows that SDB's upstream major reservoir had a stronger streamflow control
capability than the ADB's. After the collapse of the USSR, the contradiction on the question "Should water be used
for the summer irrigation water of the downstream country or the winter power generation in the upstream country?"
in both river basins has escalated but the conflict in SDB has become more and more intense and the Toktogul reservoir
operation in Kyrgyzstan has changed completely from the original natural model to a winter-release dominated mode.
However, the contribution of downstream energy supplied to the upstream country has not augmented much. This
might be due to the fact that the changes in the energy trade agreements are hard to match with the annual hydrological
cycle change. Receiving too much winter flow, the contribution of SDB's water entering the Aydar depression
increased rapidly after the disintegration and is higher than ADB. The other part of the water entering the Aydar
depression is the irrigation drainage water from collectors, which is similar to the Sarykamysh Lake in ADB. However,
during the 2005-2015 period of SDB, the sensitivity to the flow of depressions has been reduced. This may be due to
the increased water storage capacity of Kazakhstan's newly built plain reservoirs such as Koksaray, which reduces the
risk of dam failure of the Chardara reservoir located on the border of Uzbekistan and Kazakhstan. As there is no
provision in the basin water distribution agreement for the discharge of water from the Chardara reservoir to the Aydar
depression, Kazakhstan may tend to release the surplus water from the Chardara reservoir to Koksaray rather than the
Aydar depression. This will threaten the volume, water salinity, stability and fishery production (Groll et al., 2016) of
the Aydar depression in Uzbekistan and intensify the water conflict between Uzbekistan and Kazakhstan. In addition,
the contribution of some variables (such as livestock water use) has always been very low, possibly because the
livestock water consumption only accounts for a small amount of the total runoff.
**4.3 Scenario analysis and optimization of the WEFE nexus based on the BN**
Based on the Bayesian posterior probability prediction ability, we enumerated the influence of some variables on other
target nodes under different scenarios. Reducing the water volume entering depressions (Table 3) may be the most
positive and helpful to restore the ecological water entering the Aral Sea. This implies that the efficiency of salt
leaching and irrigation should be improved. It is also effective to increase the planting ratio of grain crops and reduce
cotton planting with high water consumption to ensure food security. Increasing the energy supply from upstream to
downstream area and reducing the downstream irrigation quantity per ha may also indirectly increase the ecological
water inflow to the Aral Sea. Increasing the upstream reservoir water storage and winter water release may increase
the inflow of salt water under high runoff condition. The high upstream reservoir water storage and winter water release
may indicate high runoff conditions which may also lead to an increase in the inflow of the Aral Sea. Increasing the
industrial production and animal husbandry may significantly increase GDP and livestock production. Among the
damages that need prevention, drought is the first because it has a significant effect on the desertification, soil
salinization and water mineralization.
**4.4 The multi-criteria evaluation based on the MCMC sampling of the BN**
The causal constraint of Bayesian network on the distribution range of the point set in the multi-criteria evaluation
space makes the decision makers more intuitive about the multi-dimensional uncertainty of the system (Fig. 9). We
found that the advantage of Bayesian probability theory was effectively integrated into the multi-criteria assessment.
As one of the parent nodes, the prior distribution of 'runoff' affects the probability distribution of child nodes (such as
benefit variables) through the transfer of joint probability calculations (Fig. 9). After the determination of the decision
nodes, the distribution of the point set changed (shifted from the prior joint distribution to the posterior distribution).
The distribution of comprehensive benefits under different runoffs is obviously more regular or clustered. Unlike the
independent Monte Carlo sampling of different variables which makes the distribution of point set in the multi-criteria
assessment space appear disorderly or chaotic in the previous system optimization analysis (Fig. 9), the BN-based
MCMC sampling contains the causality and dependence between sampling of different variables.
But this phenomenon varies on the specific axis of the two river basins. For example, for SDB, the degree of
cooperation (DC), which is calculated as the ratio of the upstream hydropower profit to the downstream agricultural
production, is an effective index to cluster the cases under various runoffs. In view of ADB however, the DC is not a
good index for clustering and the partial distribution pattern of the cases on the DC axis is hardly controlled by various
runoffs. This illustrates that in SDB and ADB, the relationship between the DC and the annual runoff is quite different.
The DC in SDB driven by water-energy conflict is more affected by annual runoff. When the nodes for optimization
determined ('water inflow to the depression' and 'downstream grain crop area'), in the practical decision-making, the
Pareto fronts can be solved as the optimal solution set, with no other solution than the cases which could be found
better in all three criteria in a multi-objective optimization. The solution sets under a high, medium and low runoff
could be solved separately but, in this study, we paid more attention to the uncertainty of the Pareto solutions. For
example, under a high runoff, the uncertainty of the pareto fronts of ADB is higher than the one of SDB, which shows
that if these two optimization measures are applied to ADB, the stability and robustness of the comprehensive benefits
may be lower than SDB.

**5 Discussion**

**5.1 Effectiveness and limitations of the new framework**

**5.1.1 When applied to a single river basin**

When applied to a single river basin, by measuring the involved uncertainties with joint probability, this framework can help decision makers to re-examine causal and remotely related factors that may have been overlooked before. It also helps to update their empirical knowledge of the probability distribution of some nodal variables because the previous empirical knowledge may not include the collaborative consideration of the distribution of parent nodes. Compared with process-based models, it has advantages in integrating knowledge from multi-fields and quantification of uncertainty and causality caused by data limitations and disadvantages in its ability to explain detailed processes or driving mechanisms.

The main limitations of the framework may include inappropriate selection of nodes, mismatches in the temporal and spatial representation of variables, lack of consideration of detailed causal processes and feedback causality. If the selected nodes are inappropriate, it may lead to the failure of the capture of causality. For example, it may be inappropriate for us to select the average life expectancy instead of the incidence of specific diseases caused by ecological problems such as respiratory diseases caused by sand and salt storms. The BN may not be suitable in cases that require detailed spatial and/or temporal representation (Chen and Pollino, 2012). The factors that differ from the annual scale of hydrological information may not well be modeled. For example, the changes in the energy supply from downstream to upstream might not match the variation of the annual water supply from upstream to downstream, although there is an obvious causal relation between them. In addition, the variables with cumulative values may not match the annual variation of the hydrological information. As a cumulative value, the node 'the area of the Aral Sea' is not as good as the annual water entering the Aral Sea to adapt to the annual hydrological variation and the node 'soil salinity' is also not as good as the node 'water mineralization' in order to adapt to the annual hydrological variation. Therefore, this BN trained from the yearly data may be more suitable for modeling variables that are sensitive to the annual hydrological variation, because each hydrological year is considered to be independent in this BN. The evaluation of some long-term variables may require a further integration of the process models, such as the long-term trend of soil salinization below the root zone and the long-term melting trend of the upstream glaciers with its impacts on components and spatiotemporal processes of the runoff in these river basins (Liu et al., 2011; Wang et al., 2016). The lack of a more detailed description of causality may cause some detailed but important causality to be ignored, making it difficult for us to discover the differences between river basins. Therefore, the scale to which the structure needs to be refined and when it needs to be refined are what we need to consider carefully when promoting this framework. In addition, the causal relationship between variables in the BN is unidirectional, which may make it difficult to quantify the complex interactive feedback effects (Chen and Pollino, 2012).

### 5.1.2 When applied to two or multiple river basins comparatively

In terms of comparing basins, this new BN-based framework performs well in SDB and ADB. Compared with previous comparison methods (Alcamo et al., 2003; Döll et al., 2003; Grafton et al., 2012; Immerzeel et al., 2020; Joetzjer et al., 2013; Ladson and Argent, 2002; Müller Schmied et al., 2014; Syed et al., 2005; Vetter et al., 2017; Wang et al., 2020; Zawahri, 2008), this framework is more systematic and pays more attention to the description of causality. Based on the similarity of detail causality, the comparison of the WEFE nexuses is comprehensive and meaningful in terms of historical analysis, uncertainty comparison and future system optimization. A comparative application to multiple watersheds may provide more extensive causal knowledge than only applying to a single watershed. For example, in this study, we found that care should be taken when building large reservoirs on the Panj River in the upper Amu Darya to avoid disputes over surplus water downstream caused by the release of upstream reservoirs in winter. Without the lessons of the Syr Darya, it will make it difficult to evaluate the downstream conflicts on the possible surplus water that will be caused by the further development of the Amu Darya. This may be related to the different levels of development in different river basins. Some river basins have gone through the development stage and can therefore provide lessons for the river basins that are now being rapidly developed.

Compared to process-based models, this framework quantified the actual differences between watersheds in the data-driven approach rather than in the parameter adjustment and calibration approach with the same process-based model which has shown that the issue of parameter heterogeneity is important in the global multi-watershed comparison (Alcamo et al., 2003; Döll et al., 2003; Müller Schmied et al., 2014). In the comparison of the basin-wide WEFE nexuses, we need to integrate multi-field knowledge, which may cause the problem of such parameter heterogeneity to be magnified, and the complexity of parameter adjustment will be higher. Because more parameters are included and accuracy testing is also no longer limited to the original single field. In addition, the flexibility and universality of comparison under this framework may be stronger due to the use of the form of conditional probability tables. A conditional probability table can be constructed for each watershed as a general representation of the relationship between variables, but the form of a certain equation or driving function in the process-based model may not be suitable for each watershed. In addition, in this framework, the relatively simple model structure and the use of expert knowledge enables data-limited watersheds located in developing countries to be simulated more effectively. Therefore, making the modeling effects of watersheds located in different countries comparable. In contrast, the demand for observational data for complex process-based models may be too high for data-limited watersheds located in some developing countries (Chen et al., 2017). Due to the under-refined local parameters and processes in the data-limited watersheds, comparisons based on the process-based model at the fine-scale level may be unconvincing with uncertainty.

As far as the scalability and universality of this framework are concerned, due to the similarities between the concepts of the WEFE nexus and integrated water resources management, the past water resources management studies based on BNs in some arid regions or data limited river basins (Frank et al., 2014; Keshtkar et al., 2013; Xue et al., 2017), may be able to provide additional evidence for the effectiveness of this framework. If we use this framework to compare

more river basins, we may lose a little in the details of the structure and need to consider the trade-off of structure
refinement and universality (Fig. 10). For example, comparing the Aral Sea basin with the Tarim river basin may
require removing the water-energy conflict module, because there is no energy conflict between the upper and lower
reaches in the non-transboundary Tarim river basin. However, this may also lead to deviations in the attribution of
some specific downstream water system behaviours, because the difference in upstream water-energy conflict is
ignored. In addition, the limitations of this comparison framework may include the inconsistency of network nodes
and the difference in the value range of variables. For example, the defined location and attributes of 'depressions' are
different, and the difference in the spatial extent represented by the defined 'upstream' and 'downstream' regions may
also affect the effect of comparative research. And for the same variable of different basins, the difference in the value
range and the variable status discretization operation may also bring errors to the comparison.
**5.2 The main differences between SDB and ADB concerning the WEFE nexus**
In addition to the widely recognized differences in glacier melting in high mountainous areas (Farinotti et al., 2015;
Immerzeel et al., 2020; Kraaijenbrink et al., 2017; Sorg et al., 2012), differences in interception capacity of upstream
reservoirs in these two river basins (account for 47% of total runoff of SDB and 13% of ADB) could affect the seasonal
distribution of the downstream runoff and the upper limit of the level of water-energy conflicts between the upstream
and downstream countries. In ADB, although the new Rogun dam on the Vakhsh river has been put into power in 2018,
it has a modest impact on downstream irrigation if the reservoir is operated to maximize basin-wide benefits (Jalilov
et al., 2016). We should warn that in the future some large reservoirs may be constructed on the upstream Panj river,
which would account for more than 40% of the total runoff of the Amu Darya River. If so, the water-energy conflict
between the upstream area of Tajikistan and the downstream part of Uzbekistan might escalate just like SDB. One
possible solution is to re-establish the complementary water-energy mechanism of the USSR period.
The water-energy conflicts between the upstream and downstream have gradually become accustomed, but new
conflicts and changes have been generated in the middle and lower reaches of the two rivers. In SDB, in the face of
excessive winter water discharge from Kyrgyzstan upstream, from 1991 to 2005, Kazakhstan could only release the
surplus water from the Chardara reservoir to the Aydar depression in Uzbekistan in order to reduce flooding risk.
However, after 2005, with the construction of more water conservancy projects in Kazakhstan, such as the Koksaray
reservoir built to receive surplus water from the Chardara reservoir for irrigation, the water volume of the Aydar
depression was affected. The current basin water distribution agreement does not specify the amount of water that the
Aydar depression should receive from the Chardara reservoir. If this part of the water is subtracted, the Aydar
depression can only be fed by irrigation drainage water with poor quality. These will lead to reduced water volume,
deterioration of water quality, decreased ecological stability and fishery production of the Aydar depression. Therefore,
it is necessary to pay more attention to the ecological problems of new water bodies in the water allocation of the basin,
such as determining the annual release of Kazakhstan's Chardara reservoir to Uzbekistan's Aydar depression. This is
also of reference value for Turkmenistan and Uzbekistan in the lower reaches of ADB. With the increase in population
and economic development, the contradictions in water use between downstream countries will gradually increase.
The water-food-ecology conflict between downstream countries may be a chronic problem compared to the water-
energy conflict with upstream mountainous countries.
**5.3 Other external measures**
The Bayesian network in this study was mainly based on the expert knowledge and data only within the Aral Sea basin.
It did not incorporate other potential external solutions indirectly based on the framework. But some external measures
derived from further consideration of the analysis of differences and optimization measures within the framework may
also be useful as a complement to the solutions directly based on the framework. These external measures can be
generated from the successful management experience of other river basins if more river basins are included in this
framework. After the collapse of the USSR, the decline in the agricultural demand allowed more water to flow into the
Aral Sea. But the downstream countries in the basin seemed to lack concern for ecological water demand of the Aral
Sea. The expansion of the water volume and depression area (Fig. 11) confirms this, although part of the water flow
into the depressions is necessary for the leaching of soil salt in the irrigation lands. These expanding water bodies or
wetlands could provide some ecosystem services such as fish supply. Such lower water efficiency will be challenged
in the future and saving water is the long-term solution. In addition to the repair of channels so as to reduce leakage, a
spread and large-scale drip irrigation may reduce the total water consumption by more than 30% and provide 20 to 30
km$^3$ more ecological flow for the Aral Sea. It could also lower the high-salinity groundwater levels (Fig. 11), curb the
secondary soil salinization (Zhang et al., 2014), reduce the drainage water with pesticides and salt to rivers, and reduce
diseases caused by the poor water quality downstream. The promotion of drip irrigation has been considered as useful
to improve the irrigation efficiency in other arid regions, such as the Tarim River Basin (Zhang et al., 2014) also
located in the arid region of Central Asia, of which the downstream water use efficiency has increased during recent
years after the drip irrigation promotion. Also, to reduce the water inflow to depressions may require stronger ability
to regulate runoff and improving the low efficiency of surplus water management perhaps caused by the lack of water
market regulation. Taking the Colorado River (Table 4) as an example, the construction of water conservancy facilities
in SDB and ADB could be improved. Enhancing the ability to regulate the runoff may allow a better use of the surplus
water in the high flow years but at the same time, it is necessary to avoid the upstream and downstream conflicts caused
by the new large reservoirs. Building a water market as efficient as the Colorado River in the Aral Sea Basin still seems
to have a long way to go. The Tarim River Basin has started to set prices for the irrigation water since 2003 but in most
parts of the Aral Sea Basin, the irrigation water has not been priced yet. It might depend on the economic flexibility
and a more efficient water delivery network. It is also necessary to strengthen the water-energy cooperation and to
avoid zero-sum games between the upstream and downstream countries. This is a prerequisite for an optimal
management of the Aral Sea Basin. In addition, strengthening the cooperation with the neighbouring countries, such
as Russia and China, might be helpful in terms of the water conservancy projects, energy and agricultural trade and
indirectly ease the crisis in the WEFE nexus as a result.

## 6 Conclusions

In this paper, we applied a new causal structure-based framework to compare the WEFE nexuses and applied it to SDB and ADB with the BN. The main conclusions are as follows:

(1) The new causal structure-based framework (combined with the support of actual data) is proved effective when modeling and comparing the basin-wide causal WEFE nexuses under uncertainty with a lower cost in data limited or poor gauged river basins. It may help decision support mainly in the quantification of the influence of complex causality and more remotely related variables. This systematic and causal comparison framework can be used to compare more basins based on the different levels of similarity of the causal structure.

(2) Before the collapse of the USSR, the water flow entering the Aral Sea was sensitive to the agricultural development of the two river basins. After the collapse of the USSR, its sensitivity to the water-energy conflicts between the upstream and downstream countries increased a lot. Compared with the Syr Darya, the amount of water flowing into the Aral Sea from the Amu Darya is less sensitive to the water competition between downstream summer irrigation and upstream winter hydropower partly due to the lower percentage of total runoff intercepted by upstream reservoirs. It further made the management of the surplus water in the lower reaches of SDB in winter more difficult and controversial than ADB with a large amount of water flowing into depressions outside the river and irrigation area.

(3) In the short term, reducing the water inflow to depressions and improving the planting structure prove beneficial to the Aral Sea ecology. In the long term, the construction of large reservoirs on the Panj river of the upstream ADB should be cautious so as not to get an intense water-energy conflict as SDB's. Moreover, the water-food-ecology conflict between downstream countries may escalate and turn into a long-term chronic problem such as between Kazakhstan and Uzbekistan. More attention should be paid to the reasonable ecological water consumption of new water bodies such as the Aydar-Arnasay depression in the basin-wide water allocation. It is also necessary to promote the water-saving drip irrigation and to strengthen the cooperation between internal and external countries.

## Code/Data availability

The data sources that were used in this study have been listed in the main text (Table 2). The data collected from yearbooks is available at https://doi.org/10.6084/m9.figshare.13516472 and other data is available from the links in Table 2. The Netica software used to build the Bayesian network is available from https://www.norsys.com/download.html. Intermediate data, model files and codes are available upon request from the first author H.S. (shihaiyang16@mails.ucas.ac.cn).

## Author contribution

Haiyang Shi: Conceptualization, Methodology, Software, Data, Writing. Geping Luo: Conceptualization, Supervision, Revision. Olaf Hellwich: Methodology. Hongwei Zheng: Methodology. Jie Xue: Methodology, Software. Tim Van

576 De Voorde: Supervision. Philippe De Maeyer: Supervision, Revision. Chunbo Chen, Jie Bai, Tie Liu, Shuang Liu,

577 Peng Cai, Huili He, Friday Uchenna Ochege: Data.

## Competing interests

The authors declare that they have no conflict of interest.

## Acknowledgements

We would like to thank Sabine Cnudde of Department of Geography of Ghent University for correcting English. We would also like to thank the two anonymous reviewers for their comments and suggestions which have substantially improved this paper.

## Financial support

This research has been supported by the Strategic Priority Research Programme of the Chinese Academy of Sciences (grant No. XDA20060302), the National Natural Science Foundation of China (grant No. U1803243, No. 41877012), Team project of the Chinese Academy of Sciences (grant No. 2018-YDYLTD-002), Chinese Academy of Sciences President's International Fellowship Initiative (PIFI) (grant No. 2017VCA0002), West Light Foundation of the Chinese Academy of Sciences (grant No.2018-XBQNXZ-B-011), and China's Recruitment Program of Global Experts.

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

**Figure 1 Previous comparative studies focusing on local or individual aspects (a) and more attention should be directed to**
**the identification and comparison of causality and systemicity between river basins (b).**

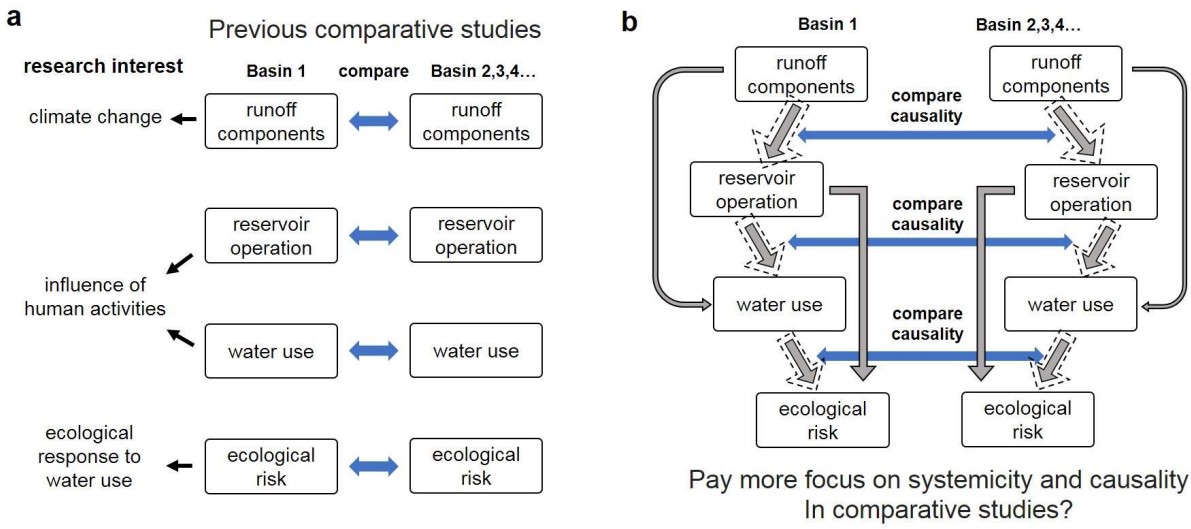


**Figure 2 The new generalized basin-wide water-energy-food-ecology nexus comparison framework based on combining the similar causal structure and data differences. The upper tree structure shows the priori classification of river basins and the arid/semi-arid branch is more subdivided. The lower left part illustrates the operation mode of the new basin comparison framework: combining the similar causal structure determined by experts and the multi-dimensional observation dataset containing differences. The red boxes marked with a, b, c, d, and e contain elements identified by the 1-12 serial number on the right that measure similarities at different levels. Number 8-10 show the different water-energy-food-ecology related nexus type adjusted according to box a, b, c, d, and e. River basins in the same red box can be compared by a specific structure of causality generated by the elements the box contains. The bottom part shows the significance of the application under this new framework.**

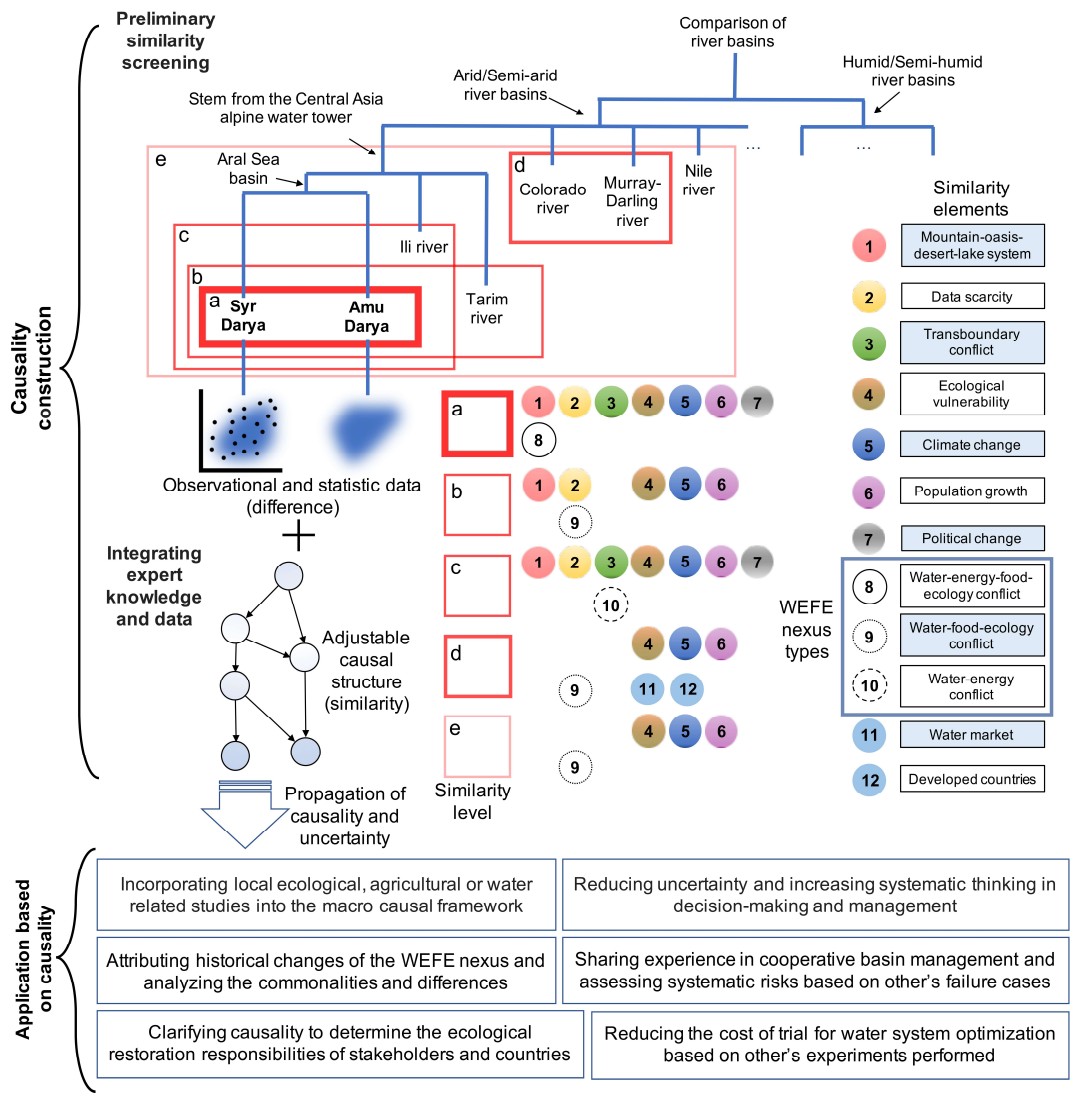

**Figure 3 Location of the Aral Sea basin and the water resources' variation. (a) shows the location of the Aral Sea Basin, the**
**two main rivers are the Syr Darya and Amu Darya. This map is made with ArcGIS and the layers come from the public**
**layers in ESRI base map and ArcGIS online. (b) demonstrates the annual runoff variation of the Syr Darya river total runoff**
**and the Amu Darya river main stream at the Atamyrat cross-section upstream the Karakum Canal.**

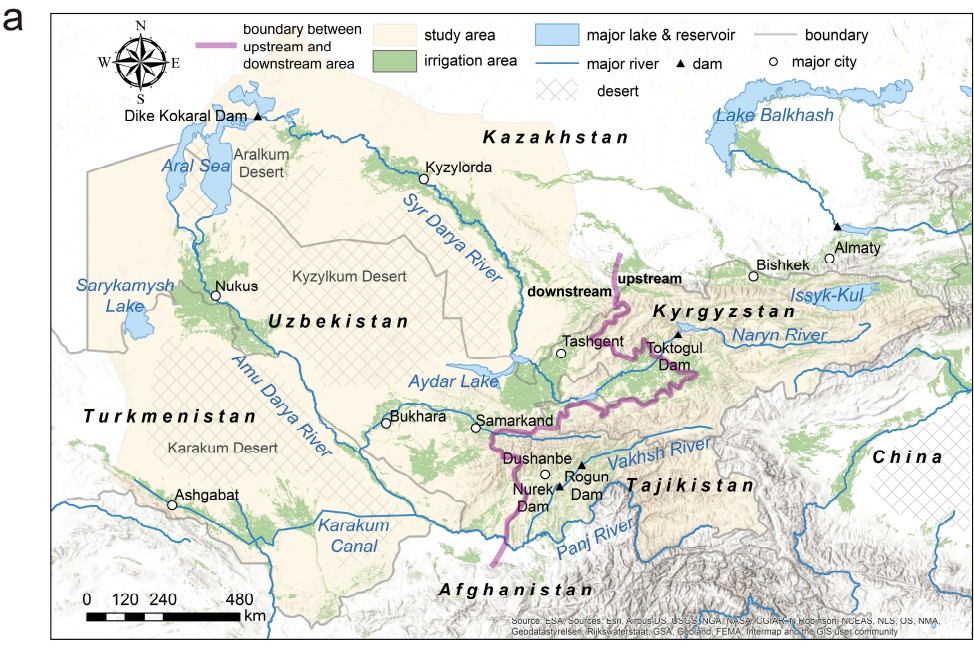

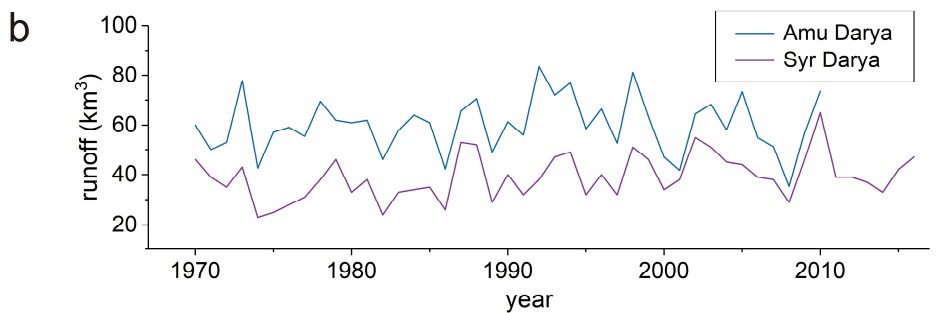


**Figure 4 The priori and general basin-wide WEFE nexus mode of SDB and ADB and its temporal change during the past 50 years (a) shows the sources of the exogenous stress on the WEFE nexus dominated by water in the Aral Sea basin. (b) illustrates the hydrologic uncertainty spread from the alpine area to the lower part through a typical 'mountain-oasis-desert-lake' system. The elements of the WEFE nexus are represented by circles in four colours and the relevant uncertainty items are tagged with these icons as a classification by respective roles in the WEFE nexus. (c) demonstrates the specific changes of the elements in the WEFE nexus during the past 50 years and the influence from the collapse of the USSR in 1991.**

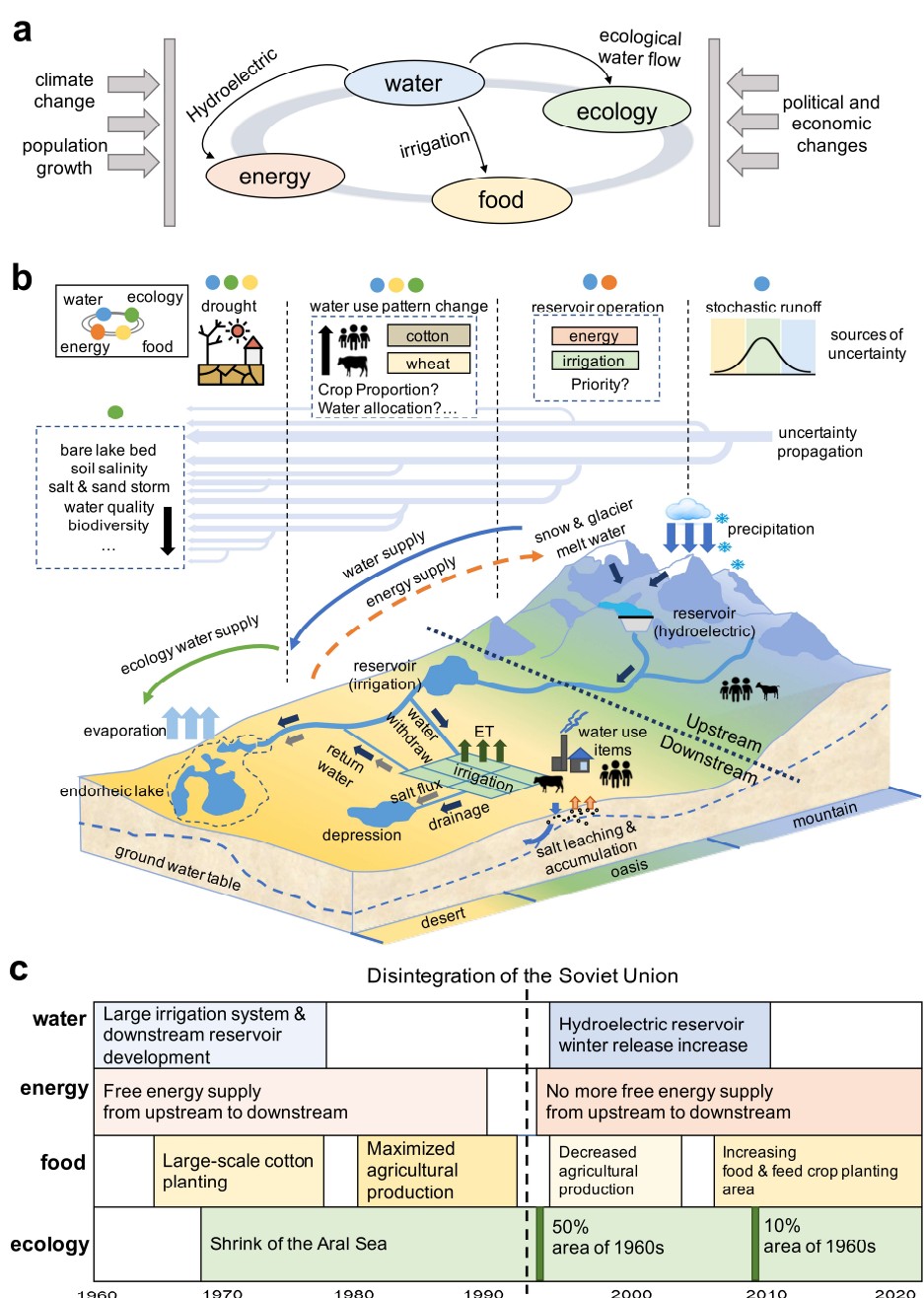




**Figure 5 Integrate expert knowledge into Bayesian networks to simulate the WEFE nexus. The geographical area is divided into the upstream, downstream region and the surrounding area of the Aral Sea. The lower part contains the factors that can be considered in the framework, and the underlined ones are actually used in this study.**

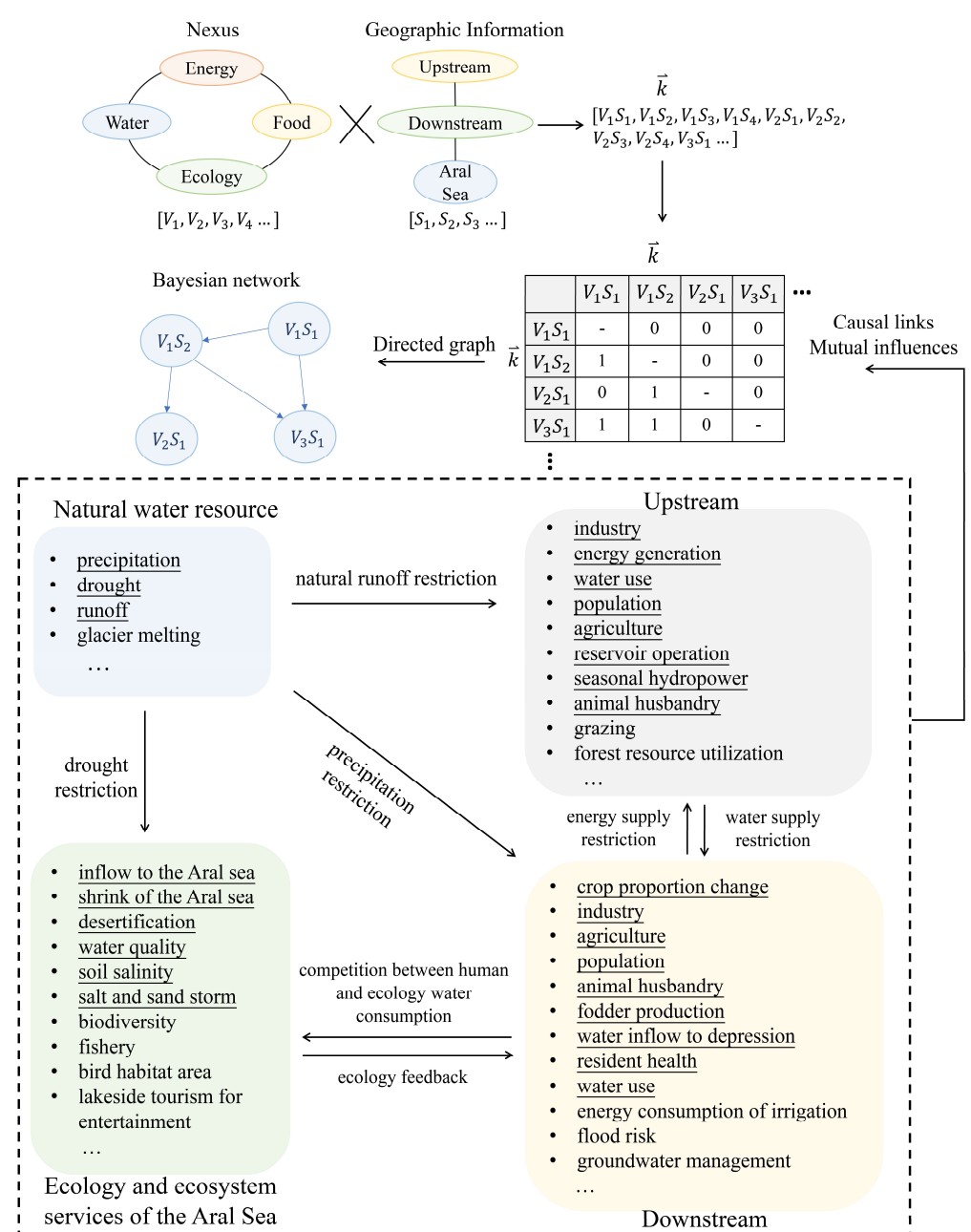




**Figure 6 The Bayesian network structure shared by ADB and SDB when simulating the water-energy-food-ecology nexus. D stands for 'downstream' and U stands for 'upstream'.**

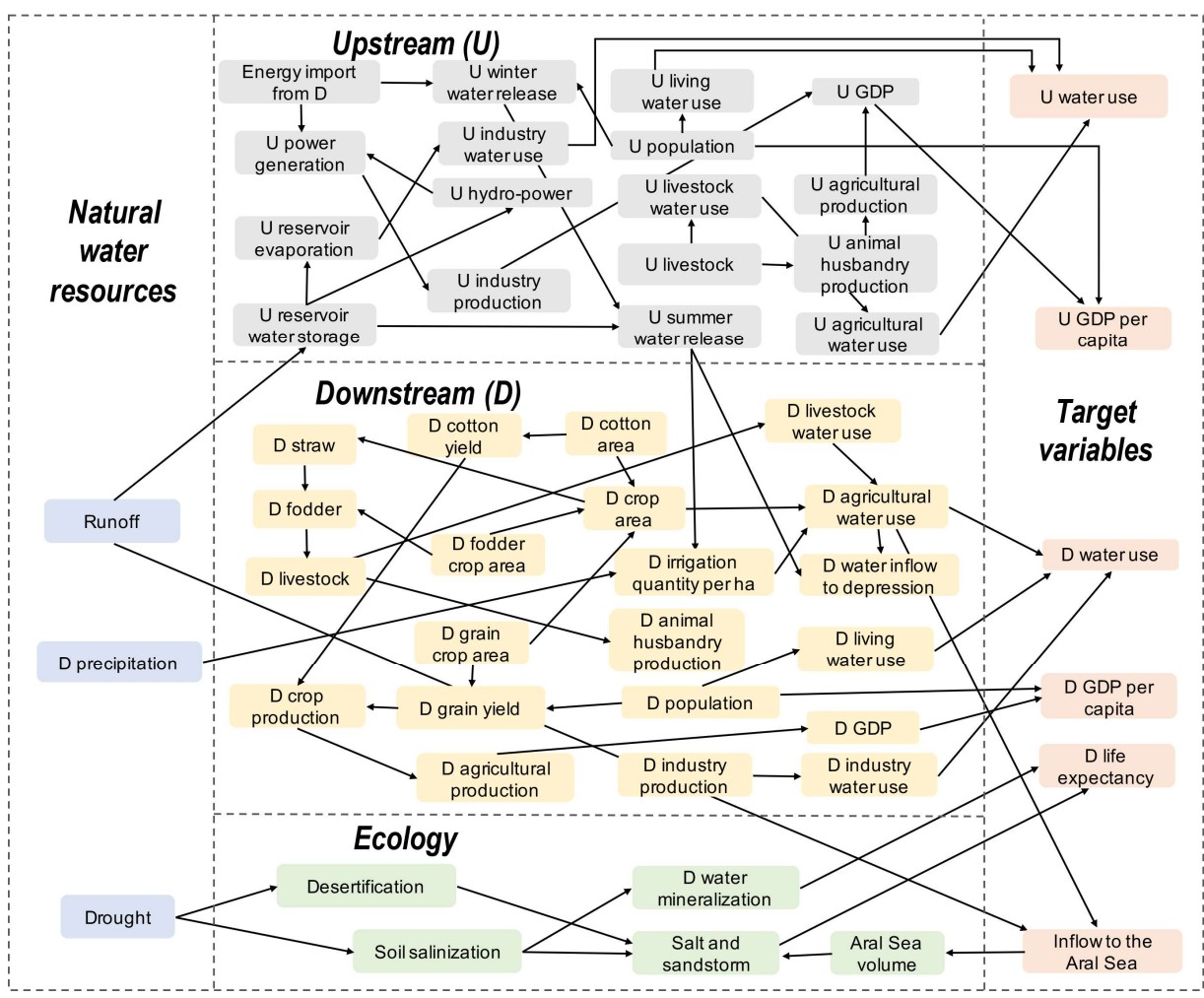




**Figure 7 Sensitivity analysis of some variables. VB stands for variance of belief and MI stands for mutual information. D**
**stands for 'downstream', correspondingly, U stands for 'upstream'.**



**Figure 8 Comparison of the sensitivity analysis of 'water inflow to the Aral Sea' node of ADB and SDB in four historical periods from 1970 to 2015. D stands for 'downstream', correspondingly, U stands for 'upstream'. VB stands for variance of belief.**

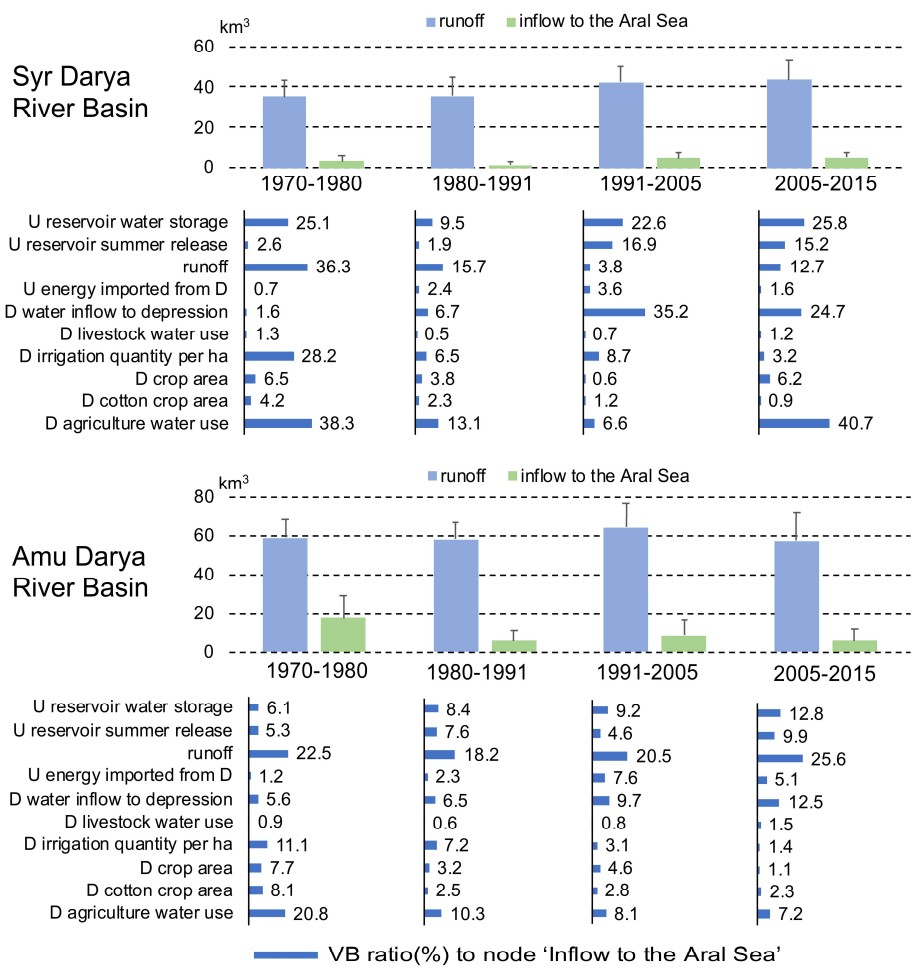



**Figure 9 Comparison of multi-criteria evaluation of SDB and ADB based on the BN causality constraint-based MCMC**
**sampling. At the top is the multi-criteria evaluation based on random sampling with no joint probability included, in the**
**middle is the multi-criteria assessment containing the BN causality constraints and at the bottom is based on the BN with**
**nodes for optimization and decision determined.**

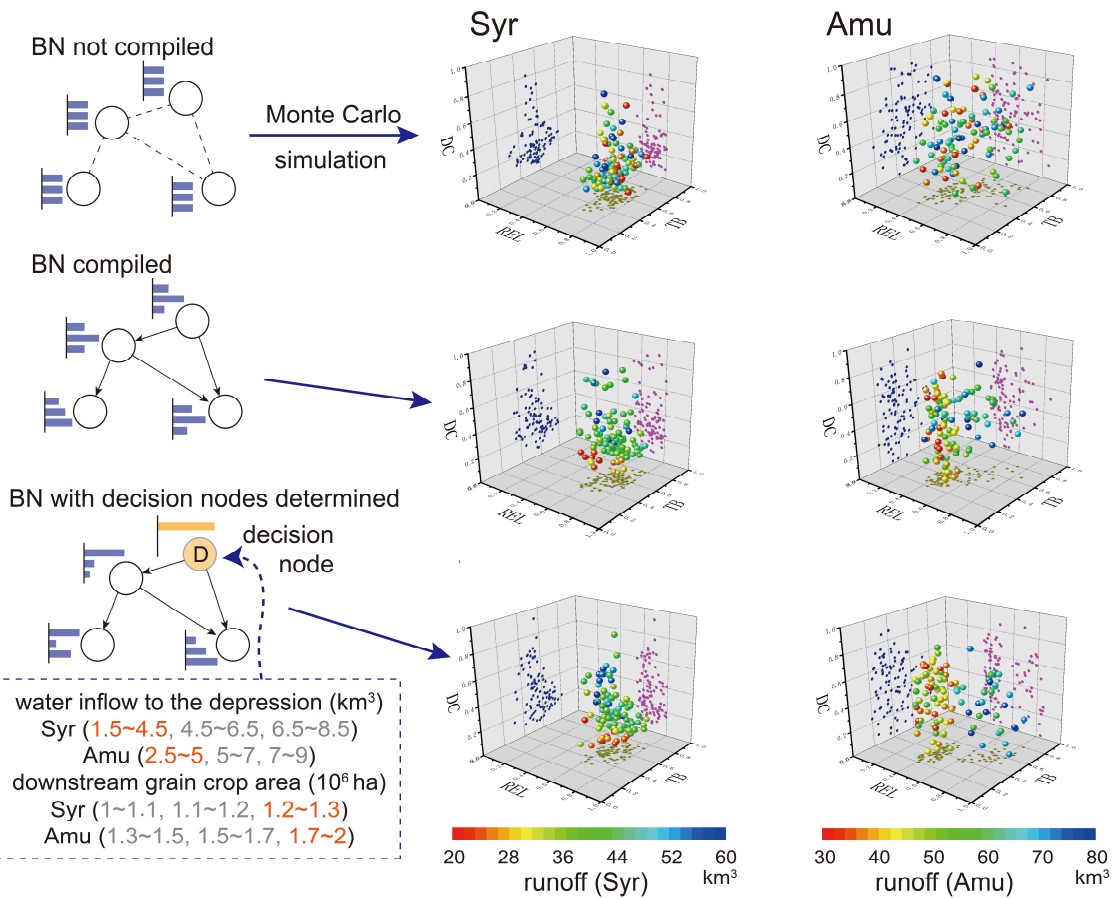




**Figure 10 The trade-off of structure refinement and universality in the new framework for comparing basin-wide water-**
**energy-food-ecology nexuses based on the adjustable causal structure.**

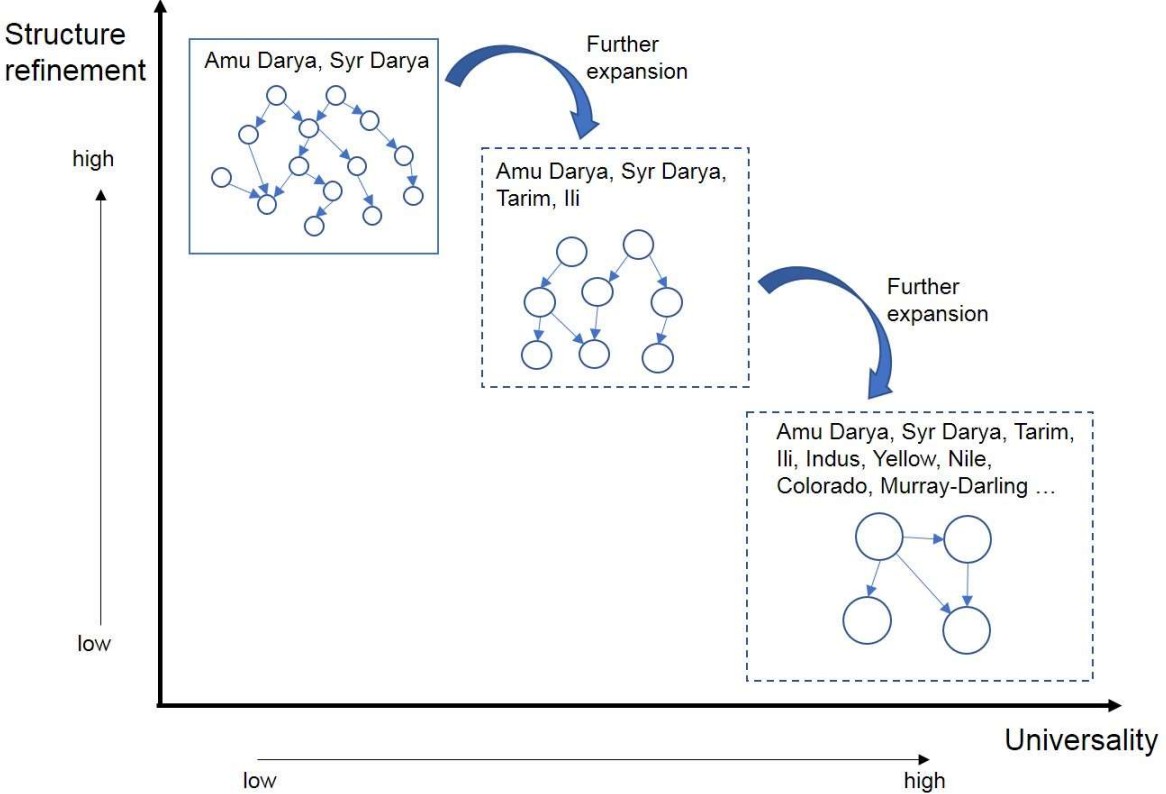




**Figure 11 The long-term inefficiency and risk of the irrigation-drainage system. (a) Changes in the surface water occurrence**
**in the Aral Sea Basin. The data and information originate from the Global Water Surface Explorer (https://global-surface-**
**water.appspot.com/) (Pekel et al., 2016). S1, A1 and A2 are examples of expanded depressions, which collected the drainage**
**and surplus water. S1 is the Aydar Lake in the Syr Darya river basin. In the Amu Darya river basin, A1 represents the**
**Sarykamysh Lake and A2 illustrates a drainage depression of the Bukhara irrigation district. (b) Salinity concentration in**
**the irrigation-drainage system of the Aral Sea Basin. The upper part stands for the salt transport and concentration at the**
**river basin scale. The lower part shows the positive effect of drip irrigation compared with flood irrigation on reducing the**
**drainage water and lowering the groundwater level to reduce the secondary salinization.**

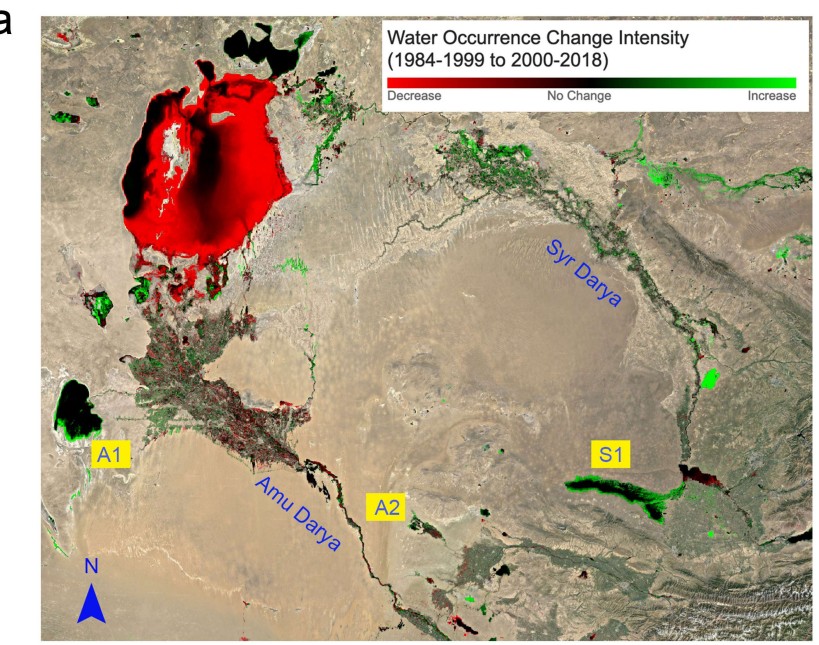

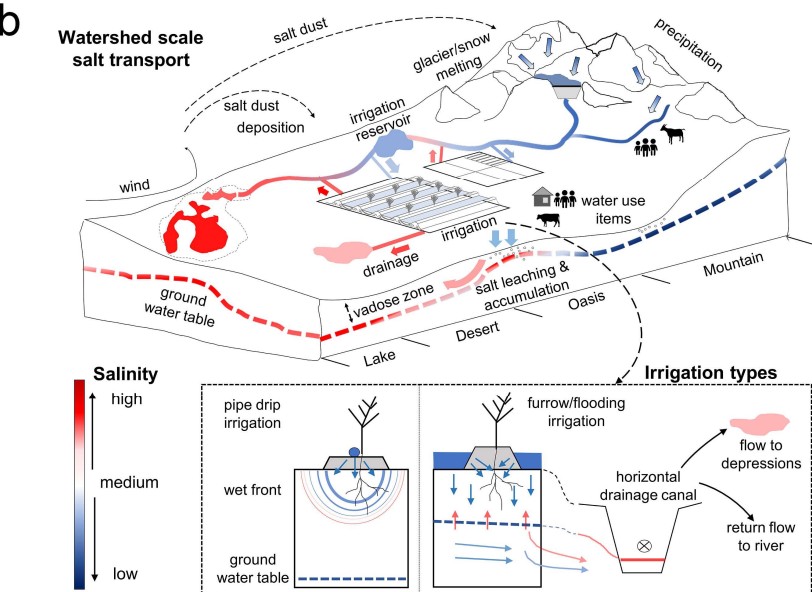



**Table 1. Discretization and description of variables**

| Variables | Status discretization | Unit | Explanation |
|---|---|---|---|
| Runoff | 280~360, 360~440, 440~650 (SDB) | $10^8$ m$^3$ | |
| | 300~500, 500~700, 700~900 (ADB) | | |
| D PDSI | -8~-4, -4~0, 0~6 (SDB) | | |
| | -8~-4, -4~0, 0~4 (ADB) | | |
| D precipitation | 170~190, 190~210, 210~230 (SDB) | mm | |
| | 80~100, 100~120, 120~150 (ADB) | | |
| U reservoir storage | 0~6, 6~12, 12~20 (SDB) | km$^3$ | Toktogul reservoir (SDB) |
| | 5~8, 8~10 10~12 (ADB) | | Nurek reservoir (ADB) |
| Outflowe of the reservoir in summer | 1800~2800, 2800~3800, 3800~4800 (SDB) | $10^6$ m$^3$ | |
| | 4000~7000, 7000~12000, 12000~15000 (ADB) | | |
| Outflow of the reservoir in winter | 3500~3800, 3800~4200, 4200~4500 (SDB) | $10^6$ m$^3$ | |
| | 2000~3000, 3000~4000, 4000~5000 (ADB) | | |
| Energy import from D | 0~1, 1~2, 2~3 (SDB) | $10^9$ m$^3$ | Natural gas export from D to U |
| | 0~0.5, 0.5~1, 1~3 (ADB) | | |
| U hydropower generation | 0.3~0.8, 0.8~1.2, 1.2~1.5 (SDB) | $10^{10}$ kW·h | |
| | 0.5~1, 1~1.4, 1.4~2 (ADB) | | |
| D cotton production | 1100~2200, 2200~3300, 3300~4400 (SDB) | $10^3$ t | |
| | 2000~2500, 2500~3000, 3000~3500 (ADB) | | |
| D cotton cropland | 700~750, 750~800, 800~850 (SDB) | $10^3$ ha | |
| | 1100~1250, 1250~1400, 1400~1600 (ADB) | | |
| D grain crop area | 1000~1100, 1100~1200, 1200~1300 (SDB) | $10^3$ ha | |
| | 1300~1500, 1500~1700, 1700~2000 (ADB) | | |
| D grain production | 1500~2500, 2500~3500, 3500~4500 (SDB) | $10^3$ t | |

| Variables | Status discretization | Unit | Explanation |
|---|---|---|---|
| | 4500~5000, 5000~5500, 5500~6500 (ADB) | | |
| Number of D livestock | 7~10, 10~13, 13~16 (SDB) | $10^6$ | cattle and sheep |
| | 10~20, 20~30, 30~40 (ADB) | | |
| D irrigation quantity per ha | 9500~10000, 10000~10500, 10500~11000 (SDB) | $m^3$/ha | |
| | 11000~13000, 13000~15000, 15000~17000 (ADB) | | |
| D water use | 33~35, 35~37, 37~40 (SDB) | $km^3$ | |
| | 45~50, 50~55, 55~60 (ADB) | | |
| Inflow to the Aral Sea | 0~4, 4~7, 7~10 (SDB) | $km^3$ | |
| | 0~7, 7~14, 14~21 (ADB) | | |
| Volume of the Aral Sea | 10~100, 100~200, 200~300 | $km^3$ | |
| Inflow to depression | 1.5~4.5, 4.5~6.5, 6.5~8.5 (SDB) | $km^3$ | Water entering the Aydar lake (SDB) |
| | 2.5~5, 5~7, 7~9 (ADB) | | Water entering the Sarykamysh lake (ADB) |
| D agricultural production | 2~4, 4~6, 6~8 (SDB) | $10^9$ US$ | |
| | 2~4, 4~7, 7~10 (ADB) | | |
| D GDP | 10~30, 30~50, 50~70 (SDB) | $10^9$ US$ | |
| | 10~40, 40~60, 60~80 (ADB) | | |
| D population | 14~16, 16~18, 18~20 (SDB) | $10^6$ | |
| | 16~18, 18~20, 20~22 (ADB) | | |
| D desertification | 14~16, 16~18, 18~20 (ADB) | $10^4$ $km^2$ | Including the Aralkum Desert |
| | 10~20, 20~30, 30~40 (SDB) | | |
| Sand and salt storm | 0~30, 30~60, 60~100 | Day per year | Frequency |

| Variables | Status discretization | Unit | Explanation |
|---|---|---|---|
| D water mineralization | 0~0.5, 0.5~1, 1~3 | g/L | Kyzylorda (SDB) |
| | | | Nukus (ADB) |
| Soil salinization | low, medium, high | | Soil salinity near Kyzylorda (SDB) |
| | | | Soil salinity near Khorezm (ADB) |
| D life expectancy | 64~66, 66~68, 68~70, 70~72 | Age | |

Note: D stands for 'downstream' and U stands for 'upstream'.

**Table 2. Data description and sources.**

| Data | Source | Description | Years 855 duration |
|------|--------|-------------|----------------|
| Discharge/run off | CA WATER info<br>http://www.cawater-info.net/water_quality_in_ca/amu_e.htm,<br>http://www.cawater-info.net/water_quality_in_ca/syr_e.htm<br>Global Runoff Data Centre (GRDC)<br>http://www.bafg.de/GRDC/EN/Home/homepage_node.html | Streamflow gauging stations, daily and yearly | 1970 to 2015 |
| Water intake and consumption | CA WATER info -  Regional Information System on Water and Land Resources in the Aral Sea Basin (CAWater-IS)<br>http://www.cawater-info.net/data_ca/?action=login<br>ICWC<br>http://sic.icwc-aral.uz/reports_e.htm, http://www.icwc-aral.uz/pdf/67-en.pdf | Province and country scale, yearly | 1970 to 2015 |
| Precipitation | National Climate Data Centre (NCDC)<br>http://www.ncdc.noaa.gov/ | Meteorological station, daily | 1970 to 2000, 2010 to 2015 |
| Palmer Drought Severity Index (PDSI) | Google Earth Engine<br>https://developers.google.com/earth-engine/datasets/catalog/IDAHO_EPSCOR_PDSI (Abatzoglou et al., 2018) | 0.04° grid, daily | 1979 to 2015 |
| Water budgets of the Aral Sea | CA WATER info -  Database of the Aral Sea<br>http://www.cawater-info.net/aral/data/index_e.htm | Annual scale | 1970 to 2015 |
| Ecological and environmental indicators | CA WATER info<br>http://www.cawater-info.net/4wwf/pdf/khamraev_e.pdf,<br>http://www.cawater-info.net/water_quality_in_ca/files/analytic_report_en.pdf,<br>http://www.cawater-info.net/water_quality_in_ca/syr_e.htm<br>Micklin P (Micklin, 1988, 2007, 2010) | Sample site scale, annual scale | 1980 to 2010 |
| Energy | CEIC<br>https://www.ceicdata.com<br>IEA<br>https://www.iea.org/data-and-statistics | Country scale, yearly | 1991 to 2015 |
| Operation of reservoirs | Siegfried T (Siegfried and Bernauer, 2007)<br>CA WATER info -  Regional Information System on Water and Land Resources in the Aral Sea Basin (CAWater-IS)<br>http://www.cawater-info.net/data_ca/?action=login,<br>http://www.cawater-info.net/projects/peer-amudarya/pdf/report_2-2_2-5_en.pdf<br>ICWC<br>http://sic.icwc-aral.uz/reports_e.htm, http://www.icwc-aral.uz/pdf/67-en.pdf | Monthly | 1974 to 2015 |
| Social economy | CA WATER info -  Regional Information System on Water and Land Resources in the Aral Sea Basin (CAWater-IS)<br>http://www.cawater-info.net/data_ca/?action=login<br>Statistical data online<br>https://stat.uz/uz, http://www.stat.kg,<br>https://data.worldbank.org.cn,<br>http://stat.gov.kz<br>FAO<br>http://www.fao.org/statistics,<br>Soviet National Economic Statistics Yearbook,<br>Commonwealth of Independent States Statistical Committee database | Province scale, yearly | 1970 to 2015 |

**856** **Table 3. Comparison of the BN-based scenario analysis of SDB and ADB**

| Target nodes | | Nodes for scenario setting | | | | | | | | | | |
|---|---|---|---|---|---|---|---|---|---|---|---|---|
| | | DR | EI | UR | WR | IQ | DG | DC | DL | UL | DI | WD |
| U energy value (high) | Syr | | +5.9 | -2.7 | +2.6 | | | | | | | |
| | Amu | | +4.4 | -1.6 | -1.2 | | | | | | | |
| D water use (low) | Syr | | | +0.2 | | +1.2 | +1.7 | | -1.6 | | -1.8 | +0.3 |
| | Amu | | | -1.1 | | -1.9 | -0.9 | | -0.6 | | -3.8 | -5.3 |
| U water use (low) | Syr | | | +2.5 | | | | | | -0.9 | | |
| | Amu | | | +0.7 | | | | | | +1.4 | | |
| D GDP (high) | Syr | | | | | | +0.6 | | +0.5 | | +4.7 | |
| | Amu | | | | | | +2.9 | | +1.4 | | +17.5 | |
| U GDP (high) | Syr | | +0.3 | | | | | | | +1.3 | | |
| | Amu | | -1.5 | | | | | | | +3.7 | | |
| D grain yield (high) | Syr | +0.3 | | | | -0.3 | +13.6 | | | | | |
| | Amu | -2.7 | | | | -2.1 | +19.3 | | | | | |
| D livestock production (high) | Syr | | | | | | | | +5.1 | | | |
| | Amu | | | | | | | | +10.3 | | | |
| Volume of the Aral Sea (high) | Syr | | | +0.6 | | | | | | | | |
| | Amu | | | +3.1 | | | | | | | | |
| Inflow to the Aral Sea (high) | Syr | | +2.6 | +3.6 | +1.3 | +2.3 | +0.5 | +2.6 | | | | +23.5 |
| | Amu | | +5.1 | +3.7 | +4.2 | +6.1 | -1.7 | +3.4 | | | | +13.2 |
| Salinization (low) | Syr | +5.5 | | | | | | | | | | |
| | Amu | +11.3 | | | | | | | | | | |
| Desertification (low) | Syr | +9.6 | | | | | | | | | | |
| | Amu | +16.2 | | | | | | | | | | |
| Water mineralization (low) | Syr | +1.3 | | | | | | | | | | |
| | Amu | +8.7 | | | | | | | | | | |
| Sand and salt storm (low) | Syr | +3.7 | | +0.8 | | | | | | | | +1.1 |
| | Amu | +13.1 | | -0.4 | | | | | | | | +0.7 |
| D life expectancy (high) | Syr | +0.2 | | | | | | | | | | |
| | Amu | -0.2 | | | | | | | | | | |

**857** Note: D stands for the downstream region and U stands for the upstream region. DR represents drought index (low), EI represents
**858** energy import from D (high), UR represents U reservoir water storage (high), WR represents U winter water release (high), DG
**859** represents D grain crop area (high), IQ represents D irrigation quantity per ha (low), DC represents D cotton crop area (low), UL
**860** represents U livestock amount (high), WD represents D water inflow to depressions (low), DI represents D industry production
**861** (high) and DL represents D livestock amount (high). The 'high' and 'low' respectively indicate the highest or lowest level of each
**862** node after discretization. The values in the table show the change of the percentage probability values of the specific states of the
**863** response nodes on the left after the 'high' or 'low' states of the upper scenario variables are determined.
**864**

865 **Table 4. Comparison of four river basins in the arid regions**

| River basin | Syr Darya | Amu Darya | Tarim river | Colorado River |
|---|---|---|---|---|
| Runoff (km$^3$) | 41 | 78 | 39 | 20 |
| Population (10$^6$) | 25 | 27 | 11 | 40 |
| Runoff / population (km$^3$/10$^6$) | 1.64 | 2.89 | 3.45 | 0.50 |
| Reservoir capacity / runoff | +++ | ++ | ++ | ++++++ |
| Hydrological observation | ++ | ++ | +++ | ++++ |
| Crop area (10$^6$ ha) | 3.3 | 4.5 | 2.8 | 1.8 |
| Runoff / crop area (km$^3$/10$^6$ ha) | 12.4 | 17.3 | 13.9 | 11.1 |
| Drip or sprinkler irrigation | + | + | +++ | +++ |
| Water market | + | + | ++ | ++++++ |
| Ecological flow | + | + | +++ | +++ |

866 Note that the number of '+' represents the values from qualitative knowledge.
867