# Peer review of "A novel causal structure-based framework for comparing basin-wide water-energy-food-ecology nexuses applied to the data-limited Amu Darya and Syr Darya river basins"

_Hydrology and Earth System Sciences, 2020_

## Referee Comment (RC1) · Anonymous Referee #1 · 26 Oct 2020

This paper presents a study of complex water-energy-food-ecology nexuses at two river basins in Central Asia. The authors used the Bayesian network to analyze the causality of the nexuses. The results indicated that the water management conflicts between downstream countries may turn into a long-term chronic problem. It is necessary to promote water conservation practices and strengthen cooperation between countries. The manuscript is on a topic of interest to the audience of HESS. I only have a few minor comments that I hope the authors could address in their revision.

Specific comments:

[Figure]

1. Section 2: Please add some references to support your proposed framework.

2. Fig. 3: Please identify the upstream and downstream areas in the map.

3. Section 4.1: This part could be elaborated to include more details. How well does the model capture the key causal links in the system? What are the limitations of the model?

4. Section 4.3: This part could also be elaborated. The authors may add discussion about the outcome of each management scenario and propose new water management strategies based on the scenario analysis results.
* * *

---

## Referee Comment (RC2) · Anonymous Referee #2 · 9 Nov 2020

Summary

The study compares watersheds by assessing causality links of different variables within the Water-energy-food-ecosystem nexus by using Bayesian Network. The framework is applied to two river basins linked to the Aral Sea, to identify factors and strategies that might explain or solve trade-offs among the different sectors and actors. The approach is of interest for the reader of HESS and Innovative. The strength of the framework is to find solutions without assuming the relationship between systems (compared to process-based models); by its design it is stakeholder and data driven.

[Figure]

The work is well illustrated with clear figures. Three main points could considerably increase the impact of the work: A – the methodology should be better described and eventually made available in order to be reproducible, B – it could be clearly stated what insights are gained from the framework and what are the limitations, C – the data used for the study should be better described and made available to the reader (as seven co-authors solely contributed on the data, it is expected to be a major contribution of this work). The following points describe the concern in detail and address a list of other minor and or important points:

A Methodology: It is not clear how this Bayesian Network is constructed, what is optimized or simulated. The authors only clearly describe 2 indicators of causality (VB and MI), and 3 performance indicators (/objectives ?): reliability, total benefits and cooperation, the rest of the framework is vague.

A1 The two following describe steps general concept of the framework, however, it is later not clearly explained how those are concretely implemented – with references to the methodology/literature. l 115. "We construct a same WEFE nexus causality structure for the river basins selected in the previous step, which can be represented by a directed graph model such as the Bayesian network" l 121. "we combine the causal structure representing expert knowledge from multiple fields with actual statistics and observation data to update the initial understanding of causality. In this way, the original qualitative causal structure is quantified by actual data, and the originally scattered actual data is closely connected by the causal structure." A2 l 196. "The responsibility for exploring the differences between the two river basins mainly relies on the continuous updates of new input cases" What are new input cases? additional data? A3 l 208. "The index variance of belief (VB) and the index mutual information (MI) based on the change of information entropy (Barton et al., 2008; Marcot, 2012) - are applied to evaluate the change in strength and uncertainty of the causal relation between the nodes." explain better what those 2 indicators mean, how they can be interpreted ? A4 l 225. " We utilized the posterior probability prediction function of BN so as to support the

decision optimization." how are "posterior probability prediction function" formulated ? – reference ? how is the "optimization" formulated – what are the variables – objective - reference? A5 In general, it is common in the HESS journal that authors if possible, provide the code, software files, so that others can use the framework. It would clearly increase the impact of the work to share the programming tools that were used (in a re-usable way).

B How does the framework guide decision making ?

B1 l 378. "In addition to the widely recognized differences in glacier melting in high mountainous areas, this study shows that the ratio of the upstream reservoir interception water to the total runoff is largely different in these two river basins" Do we need to apply the methodology to reach this conclusion ? is the data on runoff and reservoir capacity not already showing this ? B2 The rest of the promising solutions presented in sections 5.2 and 5.3 are not based on the framework, at least not to my understanding. If yes, then explained how the framework leaded to find such solutions. If not, why did the framework not help to identify those solutions? How could we improve it to do so? B3 l 444. "It might characterize the hidden uncertainty in the decision support." what hidden uncertainty has this study revealed ? B4 The limitations of the framework could be more clearly stated. In which cases will the framework fail to identify sources of problems. B5 Maybe it would be easier to understand if the authors clearly differentiated between two steps: What insights does the framework give when applying the BN to a single river basin (/case) regarding causality and management options? What insights come from the comparison of the causality links between two cases/river basins? How can one basin learn from another by looking at causality links? (what insights cannot be found by looking only at one basin) and also: How does that compare to insights found by process based model as mentioned by the authors ? what are advantages and disadvantages ?

C Data

C1 l 465. The authors state: "The data sources we used in this study have been listed in the main text. Data can also be obtained by requesting the corresponding author." The authors are referred to the guidelines of the journal regarding data: https://www.hydrology-and-earth-system-sciences.net/policies/data_policy.html In table 2, the reference to the data is not precise enough, general websites are indicated. Some links do not work, e.g. https://www.cawater-info.net needs http and not https. The data should be published in a data repository with some decent meta-data level. Seven of the co-authors contribution is solely on data, hence I expect the dataset to be a major contribution of this work, thus special attention should be payed to it.

D other comments

D1 l 22. Water-energy-food nexus or nexuses ? Inconsistent through manuscript, most literature choose nexus D2 l 22-23. the river basins did not cause the Aral sea disaster, but poor water management did. D3 l 78. many empirical parameters : give examples D4 l 89. the word "superiority" might be overclaimed. It would be also interesting to describe what type of outcomes are available from the different studies. D5 l 131. not clear what that "etc." refers to: remove. D6 l 227. "we selected the scenarios" - which scenarios are we talking about? D7 l 253. why use "international trade" market prices, what for ecosystems? D8 l 277-279. and Figure 8. "During the period 1980 - 1991, the contribution of most variables has declined, which may be related to the normalization of the maximized agricultural production, leaving only the natural runoff as the main variation contribution. " Should the sum of the contribution of all variables not match a 100%? it is stayed that the runoff becomes the main variation contribution, however also the runoff variable has a decreasing value in the VB ratio. So what is the variable that increases if all the variables showed decrease? D9 l 311. reducing flow to depressions is presented as the best solution, but in the previous section it is described as generating trade-off with other sectors and ecosystems – explain. D10 l 315. the term "positive" might be misleading, it seems drought have a positive effect on salinization, desertification in the sense "good", "desirable" . . .

---

## Editor Comment (EC1) · Murugesu Sivapalan (Editor) · 23 Dec 2020

This is a very interesting and valuable paper . The paper offers an interesting framework for comparative analysis of transboundary river water management and is therefore an important contribution to this special issue on transboundary rivers.

The authors have benefited from valuable constructive comments by two reviewers, bot of whom wanted more explanation and discussion of the methods used and the results. The authors have also responded positively to these comments and have agreed to do

the revisions requested.

Since the revisions requested and proposed are rather substantial, I will seek the opinions of the reviewers one more time. Please submit the revised version as soon as possible and I will endeavor to get it reviewed expeditiously.

———————————————

---

## Author Response (AR1)

**Response to the reviews including relevant changes made in the manuscript**

Dear Editor and Reviewers:

Thank you for the opportunity to revise our manuscript. We appreciate the careful review and constructive suggestions. It is our belief that the manuscript is substantially improved after making the suggested edits. Following this letter are the reviewer comments with our Responses (in blue) and Actions (in green), including how and where the text was modified.

Thank you for your consideration.

Sincerely,

Haiyang Shi

on behalf of authors

**Response to Anonymous Referee #1**

This paper presents a study of complex water-energy-food-ecology nexuses at two river basins in Central Asia. The authors used the Bayesian network to analyze the causality of the nexuses. The results indicated that the water management conflicts between downstream countries may turn into a long-term chronic problem. It is necessary to promote water conservation practices and strengthen cooperation between countries. The manuscript is on a topic of interest to the audience of HESS.

Response: We would like to thank the reviewer for the positive comments and the time invested to review our manuscript. The revised manuscript will follow the reviewer's recommendations.

I only have a few minor comments that I hope the authors could address in their revision.

**1** Section 2: Please add some references to support your proposed framework.

Response: Thank you for the insightful comments. We will add relevant references on how to build a Bayesian network in the fields of geography, ecology, hydrology and environment. Some previous literatures were used in the Bayesian network framework applied in this research, and they will be referenced.

Action: We heve added relevant references to support our proposed framework in Section 2. (Line 114-147)

**2** Fig. 3: Please identify the upstream and downstream areas in the map.

Response: Thank you for the insightful comments. We will identify them.

Action: We have revised the Fig. 3 of the manuscript with the upstream and downstream areas identified as follows:

[Figure]

[Figure]

**3** Section 4.1: This part could be elaborated to include more details. How well does the model capture the key causal links in the system? What are the limitations of the model?

Response: Thank you for the insightful comments. We will strengthen the analysis of the effects of the model to capture causality effects. We will also strengthen the analysis of the limitations of the framework in the discussion section. The newly added discussion content in the revised manuscript may include potential limitations caused by inappropriately selected nodes, lack of consideration of detailed causal processes, lack of expert knowledge, and low data quality/sufficiency. If a selected node variable is inappropriate, it may lead to the failure in capturing causality. For example, we used the average life expectancy instead of the incidence of specific diseases caused by ecological problems, such as respiratory diseases caused by sand and salt storms. The lack of a more detailed description of causality may cause some detailed but important causality to be ignored, making it difficult for us to discover the differences between river basins. Therefore, the scale to which the structure needs to be refined and when it needs to be refined are what we need to consider carefully when promoting this framework. Lack of expert knowledge often leads to failure when building network structures and when initializing conditional probability tables. Complex networks may often require experts or stakeholders in multiple fields, and their concerns are often different, which may cause conflicts in the setting of the network structure and the initial conditional probability table. River basins in underdeveloped areas may also lack sufficient expert knowledge due to long-term insufficient investment in local related research fields. Also weak data support (insufficient in quantity or accuracy) may weaken the effectiveness of the framework.

Action: We have revised the section 4.1:

'We input the collected data and expert knowledge into the BN and compiled it with the EM algorithm in the Netica. In this study, we selected four nodes as target variables for a sensitivity analysis (Fig. 7). We found that VB and MI have similar trends, and when VB is larger, MI is also larger. This indicates that the correlation and uncertainty between nodes are synchronized in response to changes in the parent node. The upstream power generation of the two basins is sensitive to the hydropower and imported energy. The downstream water use is more sensitive to agricultural water and living water use. The downstream agricultural production is very sensitive to crop production, animal husbandry production and soil salinization. The water inflow to the Aral Sea is sensitive to runoff, water use and reservoir operation. The ranking of these sensitivity factors matches our knowledge and experience about the Aral Sea basin well. Since the impact of the other variables in the BN gradually decreases as the number of intermediate variables increases, these sensitivity results match well with expert and stakeholder perspectives. A strong pseudo-causality was not found between two variables with no obvious prior causality. In general, the variables with a strong causality are directly (or indirectly) connected in the network. This indicates that the established priori causal structure has withstood the test of the actual data.' **(Line 315-327)**

We also modified the Section 5.1 'Effectiveness and limitions of the new framework'. We discussed the effectiveness and limitations from two aspects '5.1.1 When applied to a single river basin' and '5.1.2 When applied to two or multiple river basins comparatively'. It also includes comparisons with process-based models. **(Line 405-483)**

**4** Section 4.3: This part could also be elaborated. The authors may add discussion about the outcome of each management scenario and propose new water management strategies based on the scenario analysis results.

Response: Thank you for the insightful comments. In Section 4.3 of the original manuscript, we only

described the optimization schemes in a few scenarios. We will add discussion about more management scenarios and propose new strategies based on the scenraios in the revised manuscript.

Action: We have revised the section 4.3 of the manuscript as follows:

'Based on the Bayesian posterior probability prediction ability, we enumerated the influence of some variables on other target nodes under different scenarios. Reducing the water volume entering depressions (Table 3) may be the most positive and helpful to restore the ecological water entering the Aral Sea. This implies that the efficiency of salt leaching and irrigation should be improved. It is also effective to increase the planting ratio of grain crops and reduce cotton planting with high water consumption to ensure food security. Increasing the energy supply from upstream to downstream area and reducing the downstream irrigation quantity per ha may also indirectly increase the ecological water inflow to the Aral Sea. Increasing the upstream reservoir water storage and winter water release may increase the inflow of salt water under high runoff condition. The high upstream reservoir water storage and winter water release may indicate high runoff conditions which may also lead to an increase in the inflow of the Aral Sea. Increasing the industrial production and animal husbandry may significantly increase GDP and livestock production. Among the damages that need prevention, drought is the first because it has a significant effect on the desertification, soil salinization and water mineralization.' **(Line 367-378)**

**Response to Anonymous Referee #2**

**Summary**

The study compares watersheds by assessing causality links of different variables within the Water-energy-food-ecosystem nexus by using Bayesian Network. The framework is applied to two river basins linked to the Aral Sea, to identify factors and strategies that might explain or solve trade-offs among the different sectors and actors. The approach is of interest for the reader of HESS and Innovative. The strength of the framework is to find solutions without assuming the relationship between systems (compared to process-based models); by its design it is stakeholder and data driven. The work is well illustrated with clear figures.

Response: We would like to thank the reviewer for the positive comments and the time invested to review our manuscript. The revised manuscript will follow the reviewer's recommendations.

**A Methodology:**

It is not clear how this Bayesian Network is constructed, what is optimized or simulated. The authors only clearly describe 2 indicators of causality (VB and MI), and 3 performance indicators (/objectives ?): reliability, total benefits and cooperation, the rest of the framework is vague.

Response: Thank you for raising this point. We admit that the description of the method is not detailed enough. We will add more details about the methodology and revise this section.

Action:

In order to make the method more detailed, we have revised the manuscript as follows:

'A BN describes the joint probability distribution of the set of nodes. For a BN in which nodes represent random variables (X1,.,Xn), its joint probability distribution P(X) is given as (Pearl, 1985):

$$P(X) = P(X_1, X_2, \ldots, X_n) = \prod_{i=1}^{n} P(X_i | pa(X_i)) \qquad (1)$$

where pa(Xi) are the values of the parents of Xi and X1,.,Xn are variables in the WEFE nexus structure. Based on the expert knowledge, we initially gave values to the corresponding conditional probability table for each node of the BN.' **(Line 214)**

'Next, in order to integrate actual observations and statistical data, the expectation–maximization (EM) algorithm (Moon, 1996) function of Netica software is used to iteratively calculate the joint probability distribution of BN. In the Netica software, the "experience" variable is used to indicate the reliability of the prior knowledge, and the "degree" variable is used to indicate the training times of the observation data. By combining these two variables, we can dynamically adjust and balance the weights of prior knowledge and actual data in the probability distribution updation. In this study, we initially set "experience" <0.3 "degree" to ensure that the weight of the information represented by the actual data is sufficient.' **(Line 224-230)**

A1 The two following describe steps general concept of the framework, however, it is later not clearly explained how those are concretely implemented – with references to the methodology/literature. l 115. "We construct a same WEFE nexus causality structure for the river basins selected in the previous step, which can be represented by a directed graph model such as the Bayesian network" l 121. "we combine the causal structure representing expert knowledge from multiple fields with actual statistics and observation data to update the initial understanding of causality. In this way, the original qualitative causal

structure is quantified by actual data, and the originally scattered actual data is closely connected by the causal structure."

Response: Thank you for the insightful comments. We will add relevant references on how to build a Bayesian network in the fields of geography, ecology, hydrology and environment. Usually, expert knowledge is used for the construction of the network structure (to determine the meaning of the selected nodes and the causal logic between them) and the prior setting of the preliminary conditional probability table. In the next step, observation and statistical data are used to update the conditional probability table to get the posterior probability. We will explain in more detail how to combine expert knowledge and observed data in the revised manuscript.

Action: We have revised the the manuscript as follows:

'The conceptual structure constructed should be reviewed by a panel of experts and revised if necessary. This feedback can help identify key variables or processes that have been overlooked and correct errors in the conceptual structure. In some cases, it may be appropriate to build a conceptual structure with stakeholder groups, especially if the model will be used as a management tool and the results will affect stakeholders (Chan et al., 2010; Chen and Pollino, 2012). At the same time, the availability of actual expert knowledge and data should also be considered to avoid constructing a causal structure that is too detailed so that the available expert knowledge and data are not enough to fill it, or too rough that the causal relationship is underfitted and knowledge and data are underutilized (Chen and Pollino, 2012; Marcot et al., 2006). Including insignificant variables will increase the complexity of the network and reduce the sensitivity of the model output to important variables, unnecessarily spending extra time and effort, and will not add value to the entire model (Chen and Pollino, 2012). ' **(Line 121-131)**

'In this step, we combine the causal structure representing expert knowledge from multiple fields with actual statistics and observation data to update the initial understanding of causality (Cain, 2001; Chan et al., 2010; Chen and Pollino, 2012; Marcot et al., 2006). Expert judgment based on past observations, knowledge and experience can be used to provide an initial estimate of the probability, which can then be updated with the available observation data (Chen and Pollino, 2012). The ability to use expert opinions to parameterize the BN model is an advantage, especially for environmental systems that have little quantitative data required for statistical modeling methods (Smith et al., 2007). In this way, the conditional probability table of the original causal structure is updated with actual data, and the originally scattered actual data is closely connected by the causal structure.' **(Line 132-139)**

A2 l 196. "The responsibility for exploring the differences between the two river basins mainly relies on the continuous updates of new input cases" What are new input cases? additional data?

Response: Thank you for the insightful comments. The "new" of input cases here corresponds to the "original " of the prior probability distribution. We admit that the description here is unclear and will be revised.

Action: 'new' deleted

A3 l 208. "The index variance of belief (VB) and the index mutual information (MI) based on the change of information entropy (Barton et al., 2008; Marcot, 2012) - are applied to evaluate the change in strength and uncertainty of the causal relation between the nodes." explain better what those 2 indicators mean, how they can be interpreted ?

Response: Thank you for the insightful comments. They 'respectively' represent the reduction in variance

and entropy of the probability distribution of child nodes caused by the determination of the state of the parent nodes. As the value range of the parent node is reduced, the variance or entropy of its distribution is usually reduced. The greater the variance or entropy of the distribution of child nodes that can be further caused by this reduction, the more sensitive the child node is to the parent node which also reflects the stronger causality. We will explain this with more details in the revised manuscript.

Action: We have revised the the manuscript as follows:

'They respectively represent the reduction in variance and entropy of the probability distribution of child nodes caused by the determination of the state of the parent nodes. As the value range of the parent node is reduced, the variance or entropy of its distribution is usually reduced. The greater the variance or entropy of the distribution of child nodes that can be further caused by this reduction, the more sensitive the child node is to the parent node which also reflects the stronger causality.' **(Line 235-239)**

A4 I 225. " We utilized the posterior probability prediction function of BN so as to support the decision optimization." how are "posterior probability prediction function" formulated ? – reference ? how is the "optimization" formulated – what are the variables – objective - reference?

Response: Thank you for the insightful comments. We will revise it in the revised manuscript.

The prediction function is usually used to infer and predict how one node (D) is likely to change with the distibution of its parent node (A) determined. All nodes that have dependencies between A and D should be included in the calculation. For example, suppose we have the simple Bayesian network for discrete variables with the structure A and D are connected through a dependency of D on C ,C on B and B on A, and we can use the following formula (Heckerman and Breese, 1996) to calculate the probability of D when the state of A is given.

$$P(D|A) = \frac{P(A,D)}{P(D)} = \frac{\sum_{B,C} P(A,B,C,D)}{\sum_{A,B,C} P(A,B,C,D)} = \frac{P(A)\sum_B P(B|A)\sum_C P(C|B)P(D|C)}{\sum_A P(A)\sum_B P(A)P(B|A)\sum_C P(C|B)P(D|C)} \quad (1)$$

Parent nodes are regarded as the independent variables, child nodes are regarded as the objectives. When the state of parent node is given, the beneficial probability distribution change of the child node can be regarded as our optimization goal. We formulated a change measure ($\Delta P$) (Robertson et al., 2009) to assess the impact of a management scenario compared to a base case:

$$\Delta P_{low} = P(X_i|e)_{low} - P(X_i)_{low} \quad (2)$$
$$\Delta P_{high} = P(X_i|e)_{high} - P(X_i)_{high} \quad (3)$$

where e represents the determination of the state of the parent node (management scenario) in the form of hard evidence specifying a definite finding , $P(X_i|e)_{low}$ is the probability of the lowest state for the management scenario, $P(X_i)_{low}$ is the probability of the lowest state for the base case and $\Delta P_{low}$ is calculated as the change. The meanings of these variables are the same for the subscripts 'high'.

$$\Delta P_{low} = P(X_i|e)_{low} - P(X_i)_{low} \tag{5}$$
$$\Delta P_{high} = P(X_i|e)_{high} - P(X_i)_{high} \tag{6}$$

where e represents the determination of the state of the parent node (management scenario) in the form of hard evidence specifying a definite finding , P(Xi|e)low is the probability of the lowest state for the management scenario, P(Xi)low is the probability of the lowest state for the base case and $\Delta$Plow is calculated as the change. The meanings of these variables are the same for the subscripts 'high'.

The optimization goal of the above optimization only contains a single variable, to test whether they seemed beneficial under multiple comprehensive criteria, we selected the scenarios with a good effect ('reducing the water inflow to the depression' and 'improving the planting structure') for the multi-criteria (combination of the above single target variables) assessment.' **(Line256-277)**

A5 In general, it is common in the HESS journal that authors if possible, provide the code, software files, so that others can use the framework. It would clearly increase the impact of the work to share the programming tools that were used (in a re-usable way).

Response: Thank you for the insightful comments. We will consider making the model and code available in the "Code/Data Availablilty" section.

Action: We have revised the the manuscript as follows:

**'Code/Data availability**

The data sources that were used in this study have been listed in the main text (Table 2). The data collected from yearbooks is available at https://doi.org/10.6084/m9.figshare.13516472 and other data is available from the links in Table 2. The Netica software used to build the Bayesian network is available from https://www.norsys.com/download.html. Intermediate data, model files and codes are available upon request from the first author H.S. (shihaiyang16@mails.ucas.ac.cn).

**B How does the framework guide decision making ?**

B1 l 378. "In addition to the widely recognized differences in glacier melting in high mountainous areas, this study shows that the ratio of the upstream reservoir interception water to the total runoff is largely different in these two river basins" Do we need to apply the methodology to reach this conclusion ? is the data on runoff and reservoir capacity not already showing this ?

Response: Thank you for raising this point. We will give the values of the specific ratios, do not regard it as a conclusion, and add relevant references in the revised manuscript.

Action: We have revised the the manuscript as follows:
'In addition to the widely recognized differences in glacier melting in high mountainous areas (Farinotti et al., 2015; Immerzeel et al., 2020; Kraaijenbrink et al., 2017; Sorg et al., 2012), differences in interception capacity of upstream reservoirs in these two river basins (account for 47% of total runoff of SDB and 13% of ADB) could affect the seasonal distribution of the downstream runoff and the upper limit of the level of water-energy conflicts between the upstream and downstream countries.' **(Line 485-489)**

B2 The rest of the promising solutions presented in sections 5.2 and 5.3 are not based on the framework, at least not to my understanding. If yes, then explained how the framework leaded to find such solutions. If not, why did the framework not help to identify those solutions? How could we improve it to do so?
Response: Thank you for the insightful comments. We will revise this section. The Bayesian network in this manuscript was mainly based on the existing expert knowledge and data only within the Aral Sea basin. It did not incorporate other potential external solutions indirectly based on the framework. But we thought some external measures may also be useful as a complement to the solutions directly based on the framework. These external measures are derived from further consideration of the analysis of differences and optimization measures within the framework. For example, the promotion of drip irrigation we proposed can be seen as a further complement solution for the "reduce water inflow to depressions" in section 4.3. The discussion of the water inflow to depressions is due to the different sensitivity of the 'water inflow to the Aral Sea' to the 'water inflow to the depression' between the basins and among different periods in section 4.2. Since these factors were not considered in the network structure determined at the beginning or the degree of refinement of the structure was not sufficient to capture these factors. We have included this in the discussion section. For these promising measures in sections 5.2 and 5.3, we will try to explain their indirect connection with the framework to emphersize the logic between the sections more sufficient. For the content of measures that are not related to the framework, we will check and consider deleting them.

Action: We have revised the the manuscript as follows:
'The Bayesian network in this study was mainly based on the expert knowledge and data only within the Aral Sea basin. It did not incorporate other potential external solutions indirectly based on the framework. But some external measures derived from further consideration of the analysis of differences and optimization measures within the framework may also be useful as a complement to the solutions directly based on the framework. These external measures can be generated from the successful management experience of other river basins if more river basins are included in this framework.' **(Line 513-518)**

'Also, to reduce the water inflow to depressions may require stronger ability to regulate runoff and improving the low efficiency of surplus water management perhaps caused by the lack of water market regulation. Taking the Colorado River (Table 4) as an example, the construction of water conservancy facilities in SDB and ADB could be improved. Enhancing the ability to regulate the runoff may allow a better use of the surplus water in the high flow years but at the same time, it is necessary to avoid the upstream and downstream conflicts caused by the new large reservoirs. Building a water market as efficient as the Colorado River in the Aral Sea Basin still seems to have a long way to go.' **(Line 530-536)**

We have deleted the content in the original manuscript as follows: 'In terms of ecological restoration, since 2005, Kazakhstan's North Aral Sea Restoration Programme has separated the North and South Aral Sea by a dam so as to ensure that the water volume of the North Aral Sea is sufficient and stable (Shi et al., 2014; Singh et al., 2012), and the fisheries, biodiversity and ecosystem services are recovered in the North Aral Sea. Relatively, for the lower reaches of ADB, whether such small-scale ecological reconstruction is feasible is worthy of further assessment.'

B3 l 444. "It might characterize the hidden uncertainty in the decision support." what hidden uncertainty has this study revealed ?

Response: Thank you for the insightful comments. The framework of this article may help decision support mainly in the quantification of the influence of complex causality and more remotely related variables. It may be inappropriate and unclear to be expressed as 'hidden' here, and we will revise it to make this more clear.

Action: We have revised the the manuscript as follows:

'It may help decision support mainly in the quantification of the influence of complex causality and more remotely related variables.' **(Line 548)**

B4 The limitations of the framework could be more clearly stated. In which cases will the framework fail to identify sources of problems.

Response: Thank you for the insightful comments. We will strengthen the analysis of the limitations of the framework in Section 4.1 and discussion on it in Section 5.1. The newly added discussion content in the revised manuscript may include potential limitations caused by inappropriately selected nodes, lack of consideration of detailed causal processes, lack of expert knowledge, and low data quality/sufficiency. If the selected node variable is inappropriate, it may lead to the failure of the capture of causality. For example, we used the average life expectancy instead of the incidence of specific diseases caused by ecological problems, such as respiratory diseases caused by sand and salt storms. The lack of a more detailed description of causality may cause some detailed but important causality to be ignored, making it difficult for us to discover the differences between river basins. Therefore, the scale to which the structure needs to be refined and when it needs to be refined are what we need to consider carefully when promoting this framework. Lack of expert knowledge often leads to failure when building network structures and when initializing conditional probability tables. Complex networks may often require experts or stakeholders in multiple fields, and their concerns are often different, which may cause conflicts in the setting of the network structure and the initial conditional probability table. River basins in underdeveloped areas may also lack sufficient expert knowledge due to long-term insufficient investment in local related research fields. And weak data support (insufficient in quantity or accuracy) may also weaken the effectiveness of the framework.

Action: We have revised the the manuscript as follows:

'The main limitations of the framework may include inappropriate selection of nodes, mismatches in the temporal and spatial representation of variables, lack of consideration of detailed causal processes and feedback causality. If the selected nodes are inappropriate, it may lead to the failure of the capture of causality. For example, it may be inappropriate for us to select the average life expectancy instead of the

incidence of specific diseases caused by ecological problems such as respiratory diseases caused by sand and salt storms. The BN may not be suitable in cases that require detailed spatial and/or temporal representation (Chen and Pollino, 2012). The factors that differ from the annual scale of hydrological information may not well be modeled. For example, the changes in the energy supply from downstream to upstream might not match the variation of the annual water supply from upstream to downstream, although there is an obvious causal relation between them. In addition, the variables with cumulative values may not match the annual variation of the hydrological information. As a cumulative value, the node 'the area of the Aral Sea' is not as good as the annual water entering the Aral Sea to adapt to the annual hydrological variation and the node 'soil salinity' is also not as good as the node 'water mineralization' in order to adapt to the annual hydrological variation. Therefore, this BN trained from the yearly data may be more suitable for modeling variables that are sensitive to the annual hydrological variation, because each hydrological year is considered to be independent in this BN. The evaluation of some long-term variables may require a further integration of the process models, such as the long-term trend of soil salinization below the root zone and the long-term melting trend of the upstream glaciers with its impacts on components and spatiotemporal processes of the runoff in these river basins (Liu et al., 2011; Wang et al., 2016). The lack of a more detailed description of causality may cause some detailed but important causality to be ignored, making it difficult for us to discover the differences between river basins. Therefore, the scale to which the structure needs to be refined and when it needs to be refined are what we need to consider carefully when promoting this framework. In addition, the causal relationship between variables in the BN is unidirectional, which may make it difficult to quantify the complex interactive feedback effects (Chen and Pollino, 2012).' **(Line 415-436)**

B5 Maybe it would be easier to understand if the authors clearly differentiated between two steps: What insights does the framework give when applying the BN to a single river basin (/case) regarding causality and management options? What insights come from the comparison of the causality links between two cases/river basins? How can one basin learn from another by looking at causality links? (what insights cannot be found by looking only at one basin) and also: How does that compare to insights found by process based model as mentioned by the authors ? what are advantages and disadvantages ?
Response: Thank you for the insightful comments. This is a good suggestion. We will rearrange the results and discussion sections in the order of the questions listed by the reviewer.

When applied only to a single basin, this framework can help decision makers to re-examine causal and remotely related factors that may have been overlooked before. It also helps to update their empirical knowledge of the probability distribution of some node variables because the previous empirical knowledge may not include the collaborative consideration of the distribution of parent nodes. Compared with process-based models, it may have advantages in the quantification of uncertainty and causality when data-limited and disadvantages in its ability to explain detailed processes or driving mechanisms.

When applied to two or more basins, comparing the differences in causal links between different river basins can help local decision makers visualize the possible benefits or risks of new decisions because other river basins may have experienced similar decisions during the development process. We have already pointed out in the original manuscript that care should be taken when building large reservoirs on the Panj River in the upper Amu Darya to avoid disputes over surplus water downstream caused by the release of upstream reservoirs in winter. Without this experience of the Syr Darya, it will make it difficult

to evaluate the downstream conflicts on the possible surplus water that will be caused by the further development of the Amu Darya. Compared with applying the same process-based model in multiple watersheds, it may have advantages in simultaneously and dynamically showing the various causal relationships based on various combined conditions. The new framework may also be able to avoid errors caused by using different parameter groups when applied to two or more basins in the process-based framework because it entrusts the task of discovering differences between river basins to the actual observational data instead of pre-setting or adjusting different parameters of the driving functions in the process-based model. This part will be elaborated in the revised manuscript.

Action: We have revised the the manuscript as follows (differentiated between two steps):
We modified the Section 5.1 'Effectiveness and limitions of the new framework'. We discussed the effectiveness and limitations from two aspects '5.1.1 When applied to a single river basin' and '5.1.2 When applied to two or multiple river basins comparatively'. It also includes comparisons with process-based models. **(Line 405-483)**

**C Data**
C1 l 465. The authors state: "The data sources we used in this study have been listed in the main text. Data can also be obtained by requesting the corresponding author." The authors are referred to the guidelines of the journal regarding data: https://www.hydrology-and-earth-system-sciences.net/policies/data_policy.html In table 2, the reference to the data is not precise enough, general websites are indicated. Some links do not work, e.g. https://www.cawater-info.net needs http and not https. The data should be published in a data repository with some decent meta-data level. Seven of the co-authors contribution is solely on data, hence I expect the dataset to be a major contribution of this work, thus special attention should be payed to it.
Response: Thank you for the insightful comments. We will make the data easier to access by checking and modifying data source links or publishing it in a data repository.

Action: We have revised the Table 2 with data source links checked and modified. We also modified the Code/Data availability as follows:

**'Code/Data availability**
The data sources that were used in this study have been listed in the main text (Table 2). The data collected from yearbooks is available at https://doi.org/10.6084/m9.figshare.13516472 and other data is available from the links in Table 2. The Netica software used to build the Bayesian network is available from https://www.norsys.com/download.html. Intermediate data, model files and codes are available upon request from the first author H.S. (shihaiyang16@mails.ucas.ac.cn).

**D other comments**
D1 l 22. Water-energy-food nexus or nexuses ? Inconsistent through manuscript, most literature choose nexus.
Response: Thank you for the insightful comments. Our understanding is that 'es' can be added when representing nexuses in multiple basins. We will check and make it consistent.
Action: Checked

D2 l 22-23. the river basins did not cause the Aral sea disaster, but poor water management did.

Response: Thank you for the insightful comments. We will revise it.

Action: Modified.

D3 l 78. many empirical parameters : give examples

Response: Thank you for the insightful comments. We will revise it. In the original manuscript we described it (line 78): 'However, in order to parameterize these models, we found that many empirical parameters or factors need to be set (Feng et al., 2016; Ravar et al., 2020), which could mask the shortcomings of an insufficient understanding of uncertain and complex processes.' For example, in these two articles, empirical coefficients are used to determine the conversion coefficient of electricity demand for water pumping from different depths, energy demand coefficients of various water sectors (Ravar et al., 2020) and driving functions of water supply, power generation, hydro-ecology (Feng et al., 2016).

Response: Thank you for the insightful comments. We will revise it.

Action: Replaced with 'advantage'

D5 l 131. not clear what that "etc." refers to: remove.

Response: Thank you for the insightful comments. We will remove it.

Action: Removed.

D6 l 227. "we selected the scenarios" - which scenarios are we talking about?

Response: Thank you for the insightful comments. It refers to reducing the water inflow to the depression and improving the planting structure. We will revise it.

Action: We have revised it.

'The optimization goal of the above optimization only contains a single variable, to test whether they

seemed beneficial under multiple comprehensive criteria, we selected the scenarios with a good effect ('reducing the water inflow to the depression' and 'improving the planting structure') for the multi-criteria (combination of the above single target variables) assessment.' **(Line 275-276)**

D7 I 253. why use "international trade" market prices, what for ecosystems?

Response: Thank you for the insightful comments. We use "international trade" market prices here because when it comes to cross-border cooperative management, different types of benefits (such as upstream hydropower and downstream agricultural products) may need to be weighted and summed. It may be more reasonable to use the universal price of various benefits in the international market to determine the weight. The value of ecological flow can be replaced by calculating the value of the ecosystem services it provides. In this manuscript, we did not use the actual price in this step because actual prices are constantly changing, and there is no uniform method for calculating the value of ecosystem services to determine the weight of ecological flows. But in the practical management, unique values should be given to determine the weight. As a simplified calculation, we normalized the three indicators to 0-1 and sum them with equal weights. We will revise this section to illustrate this clearly.

Action: We have revised the the manuscript as follows:

'$P_a$, $P_h$ and $P_e$ are the prices or weights which can be adjusted according to the actual market price in the international trade when it comes to cross-border cooperative management in which different types of benefits (such as upstream hydropower and downstream agricultural products) may need to be weighted and summed. It may be more reasonable to use the universal price of various benefits in the international market to determine the weight. The value of ecological flow can be calculated as the value of the ecosystem services it provides. As a simplified calculation, we normalized the three indicators to 0-1 and sum them with equal weights.' **(Line 300-306)**

D8 I 277-279. and Figure 8. "During the period 1980 - 1991, the contribution of most variables has declined, which may be related to the normalization of the maximized agricultural production, leaving only the natural runoff as the main variation contribution. " Should the sum of the contribution of all variables not match a 100%? it is stayed that the runoff becomes the main variation contribution, however also the runoff variable has a decreasing value in the VB ratio. So what is the variable that increases if all the variables showed decrease?

Response: Thank you for the insightful comments. It is normal that the sum of the VB ratio is not 100%. It reflects the sensitivity of the target variable to different parent variables, so the reasonable range of each VB ratio can be 0-1. The decline in the VB ratios of this period may be due to the fact that the inflow to the Aral Sea has been steadily very low during this period.

D9 I 311. reducing flow to depressions is presented as the best solution, but in the previous section it is described as generating trade-off with other sectors and ecosystems – explain.

Response: Thank you for the insightful comments. In the previous section, it specifically refers to the benefits to the lake ecosystem of the depression. In section 4.3, the optimization goal here is focused the inflow to the Aral Sea. We will revise it to make the statement more clear.

Action: We have revised the the manuscript as follows:

'EB is the benefit of downstream ecological flow **entering the Aral Sea** calculated as a linear function of WECO in this paper' **(Line 298)**

D10 l 315. the term "positive" might be misleading, it seems drought have a positive effect on salinization, desertification in the sense "good", "desirable" . . .
Response: Thank you for raising this point. We will revise it.
Action: We have delected the word 'positive'.

[revised manuscript text omitted]